# Reweighting Improves Conditional Risk Bounds

**Yikai Zhang** [*]
*Machine Learning Research, Morgan Stanley*

*Yikai.Zhang@morganstanley.com*

**Jiahe Lin** [*]
*Machine Learning Research, Morgan Stanley*

*Jiahe.Lin@morganstanley.com*

**Fengpei Li**
*Machine Learning Research, Morgan Stanley*

*Fengpei.Li@morganstanley.com*

**Songzhu Zheng**
*Machine Learning Research, Morgan Stanley*

*Songzhu.Zheng@morganstanley.com*

**Anant Raj**
*Machine Learning Research, Morgan Stanley*

*Anant.Raj@morganstanley.com*

**Anderson Schneider**
*Machine Learning Research, Morgan Stanley*

*Anderson.Schneider@morganstanley.com*

**Yuriy Nevmyvaka**
*Machine Learning Research, Morgan Stanley*

*Yuriy.Nevmyvaka@morganstanley.com*

**Reviewed on OpenReview:** *https://openreview.net/forum?id=MvYddudHuE*

## Abstract

In this work, we study the weighted empirical risk minimization (weighted ERM) schema, in which an additional data-dependent weight function is incorporated when the empirical risk function is being minimized. We show that under a general "balanceable" Bernstein condition, one can design a weighted ERM estimator to achieve superior performance in certain sub-regions over the one obtained from standard ERM, and the superiority manifests itself through a data-dependent constant term in the error bound. These sub-regions correspond to large-margin ones in classification settings and low-variance ones in heteroscedastic regression settings, respectively. Our findings are supported by evidence from synthetic data experiments.

## 1 Introduction

The empirical risk minimization (ERM) schema plays an important role in tackling modern machine learning tasks. Given a set of samples $S_n \triangleq \{z_i \triangleq (x_i, y_i)\}_{i=1}^n$ that are often assumed i.i.d., ERM estimates the target hypothesis $f$ within some hypothesis class $\mathcal{F}$ by minimizing the *empirical risk* based on $S_n$, that is, $\widehat{f} \triangleq \arg\min_{f \in \mathcal{F}} \frac{1}{n} \sum_{z_i \in S_n} \ell(f, z_i)$, where $\ell$ is some loss function. The method is widely adopted due to its ease of use and generalization power.

Let $z \triangleq (x, y)$ and $f^* \triangleq \arg\min_{f \in \mathcal{F}} \mathbb{E}[\ell(f, z)]$, the *excess risk* is defined as:

$$\text{Excess Risk} \triangleq \mathbb{E}\ell(\widehat{f}, z) - \mathbb{E}\ell(f^*, z).$$

Under suitable conditions, ERM achieves low excess risk with high probability. Specifically, it is known to achieve a minimax optimal rate of $O(n^{-1/2})$ for learning tasks under general settings (Vapnik & Chervonenkis,

---

Correspondence to: Yikai Zhang. ⟨zhangyikai91@gmail.com⟩; ∗: equal contribution

1974; Talagrand, 1994; Boucheron et al., 2005), and an $O(n^{-1})$ fast rate in more restrictive ones (Koltchinskii & Panchenko, 2000; Mendelson, 2002a; Bartlett et al., 2005; Boucheron et al., 2005), where matching lower bounds have also been established (Massart & Nédélec, 2006; Zhivotovskiy & Hanneke, 2018).

More recently, alternative risk minimization schema that outperforms ERM on certain sub-regions has been proposed in several works. Notably, Namkoong & Duchi (2017) introduces a distributionally robust optimization schema that is provably superior to ERM in scenarios where ERM is susceptible to noise; e.g., when the loss is piecewise linear and the thus under/over-estimation of the slope has a sizable impact to the solution. In Xu & Zeevi (2024), a carefully designed moment penalization function is introduced and it achieves a superior rate than ERM within the large-variance hypothesis class. Bousquet & Zhivotovskiy (2021) introduces a rejection option to achieve fast rate of convergence in the absence of margin conditions; note that these conditions are typically required for establishing fast rate for ERM in settings absent of the rejection option. The aforementioned studies point to different directions of improvement over the standard ERM.

In this work, we leverage the problem-dependent structure to improve upon ERM, which provably outperforms in "high confidence" regions. Specifically, instead of minimizing directly the empirical loss, a re-weighted version based on some weight function $\omega(\cdot)$ is considered, which leads to an estimator that minimizes the *weighted empirical risk* of the following form:

$$\widehat{f}_{\mathrm{wERM}} \triangleq \arg\min_{f \in \mathcal{F}} \frac{1}{n} \sum_{i=1}^{n} \omega(\boldsymbol{x}_i)\ell(f; \boldsymbol{z}_i). \tag{1}$$

We show that such an estimator improves the *conditional excessive risk*, given by

$$\mathbb{E}_{\boldsymbol{z}}\Big(\ell(\widehat{f}; \boldsymbol{z}) - \ell(f^*; \boldsymbol{z})|\{\omega(\boldsymbol{x}) > c\}\Big),$$

with the conditioning region characterized by a level set associated with the data-dependent weight. Such an improvement can be of interest to practitioners in settings akin to those considered in the selective learning literature, where the focus is on model outcomes associated with selected sub-regions.

Our approach is inspired by an analysis of a Bernstein-type condition of the form $\mathbf{Var}[h(\boldsymbol{z})] \leq B\mathbb{E}[h(\boldsymbol{z})]$ for all $h$ in some function class $\mathcal{H}$ (e.g., Bartlett & Mendelson, 2006), which is often required for local Rademacher complexity-based analysis (Bartlett et al., 2005). In particular, one way to derive the $O(n^{-1})$ fast rate is to leverage some variance-aware inequalities such as the Bernstein inequality (Boucheron et al., 2005) and Talagrand's inequality (Talagrand, 1994; Bartlett et al., 2005), which gives rise to intermediate results of the form:

$$\mathbb{E}_{\boldsymbol{z}}[\ell(\widehat{f}, \boldsymbol{z}) - \ell(f^*, \boldsymbol{z})] \leq \frac{1}{n} \sum_{\boldsymbol{z}_i \in S_n} \{\ell(\widehat{f}, \boldsymbol{z}_i) - \ell(f^*, \boldsymbol{z}_i)\} + a\sqrt{\frac{\mathbf{Var}[\ell(\widehat{f}, \boldsymbol{z}) - \ell(f^*, \boldsymbol{z})]}{n}} + \frac{b}{n}, \tag{2}$$

for some problem-dependent constant $a$ and $b$. Under a Bernstein-type condition, the variance term on the right-hand side of inequality (2) is replaced by $B\mathbb{E}_{\boldsymbol{z}}[\ell(\widehat{f}, \boldsymbol{z}) - \ell(f^*, \boldsymbol{z})]$. The multiplier $B$ is usually chosen in a conservative way; for example, it is chosen as the inverse of the minimum margin present in the Massart noise condition (Massart & Nédélec, 2006) for classification problems or the uniform upper bound of regression loss in bounded regression settings (Boucheron et al., 2005; Bartlett et al., 2005).

On the other hand, a vanilla conservative choice of $B$ in $\mathbf{Var}[h] \leq B\mathbb{E}[h]$ is not necessarily optimal. Under the weighted ERM schema where a carefully designed weighted empirical risk in (1) is considered, the Bernstein condition could be "balanced" as: $\mathbf{Var}[\omega(\boldsymbol{x})h(\boldsymbol{x})] \leq \mathbb{E}[\omega(\boldsymbol{x})h(\boldsymbol{x})]$. Crucially, the constant $B$ can be eliminated, which subsequently leads to a tighter bound (up to some problem-dependent constant) in *excess risk* in certain sub-regions. Specifically, within the context of classification and heteroscedastic bounded regression settings, these regions can be characterized as the large-margin and the low-variance ones, respectively. More generally, the conclusion is applicable to learning tasks whose loss function satisfies a *Balanceable Bernstein Condition* when the weight function can be designed accordingly.

**Contribution** The major contribution of this paper can be summarized as follows.

- We consider the weighted ERM schema and investigate its theoretical properties; in particular, we show that it admits a tighter bound on excessive risk up to some problem-dependent constant, conditional on specific sub-regions. The enhanced conditional excessive risk bound also implies an improved unconditional excessive risk bound, provided that the noise satisfies certain conditions. See Table 1 for a comparison.

- Empirically, we demonstrate that with a properly designed weight function, one can achieve superior performance in selected regions, respectively for heteroscedastic regression and classification tasks.

- The proofs of the theorems rely on an alternative version of Theorem 3.3 from Bartlett et al. (2005), which is based on a relaxed Bernstein-type condition that allows for an $\varepsilon$ additive error. The exact arguments and strategies adopted are technical tools that are of independent interest to the community.

- As a by-product, the $O(1/n)$ fast rate for learning the variance function $\sigma^2(\boldsymbol{x})$ derived in the regression example (Theorem 4.6) improves over existing results in Zhang et al. (2023) that has an $O(1/\sqrt{n})$ rate.

| | ERM | | Weighted-ERM | |
|---|---|---|---|---|
| **Classification** | | | | |
| risk function | $\ell(f;\boldsymbol{z}) \triangleq \mathbb{1}\{f(\boldsymbol{x}) \neq y\}$ | | $\omega \cdot \ell(f;\boldsymbol{z}) \triangleq \widehat{\omega}(\boldsymbol{x})\mathbb{1}\{f(\boldsymbol{x}) \neq y\}$ | |
| general Setting | $\mathbb{E}_{\boldsymbol{z}}[\ell(\widehat{f};\boldsymbol{z}) - \ell(f^*;\boldsymbol{z})] \leq \frac{\varepsilon}{\gamma}$ | $\diamond$ | $\mathbb{E}_{\boldsymbol{z}}[\ell(\widehat{f};\boldsymbol{z}) - \ell(f^*;\boldsymbol{z})] \leq \frac{\varepsilon}{\gamma}$ | $\circ$ |
| large margin region $\mathbf{R} \triangleq \{\omega^*(\boldsymbol{x}) > c\}$ | $\mathbb{E}_{\boldsymbol{z}}[\ell(\widehat{f};\boldsymbol{z}) - \ell(f^*;\boldsymbol{z})\|\mathbf{R}] \leq \frac{\varepsilon}{\gamma}\frac{1}{\mathbb{P}(\mathbf{R})}$ | $\diamond$ | $\mathbb{E}_{\boldsymbol{z}}[\ell(\widehat{f};\boldsymbol{z}) - \ell(f^*;\boldsymbol{z})\|\mathbf{R}] \leq \frac{\varepsilon}{c}\frac{1}{\mathbb{P}(\mathbf{R})}$ | $\circ$ |
| low-margin diminishing $(1 - \mathbb{P}(\mathbf{R}))c^2 \leq \varepsilon$ | $\mathbb{E}_{\boldsymbol{z}}[\ell(\widehat{f};\boldsymbol{z}) - \ell(f^*;\boldsymbol{z})] \leq \frac{\varepsilon}{\gamma}$ | $\diamond$ | $\mathbb{E}_{\boldsymbol{z}}[\ell(\widehat{f};\boldsymbol{z}) - \ell(f^*;\boldsymbol{z})] \leq \frac{\varepsilon}{c}$ | $\circ$ |
| **Regression** | | | | |
| risk function | $\ell(f;\boldsymbol{z}) \triangleq (f(\boldsymbol{x}) - y)^2$ | | $\omega \cdot \ell(f;\boldsymbol{z}) \triangleq (f(\boldsymbol{x}) - y)^2/\widehat{\sigma^2}(\boldsymbol{x})$ | |
| general Setting | $\mathbb{E}_{\boldsymbol{z}}[\ell(\widehat{f};\boldsymbol{z}) - \ell(f^*;\boldsymbol{z})] \leq \varepsilon/\gamma^2$ | $\diamond$ | $\mathbb{E}_{\boldsymbol{z}}[\ell(\widehat{f};\boldsymbol{z}) - \ell(f^*;\boldsymbol{z})] \leq \varepsilon/\gamma^2$ | $\circ$ |
| low variance region $\mathbf{R} \triangleq \{\sigma^{2*}(\boldsymbol{x}) < 1/c\}$ | $\mathbb{E}_{\boldsymbol{z}}[\ell(\widehat{f};\boldsymbol{z}) - \ell(f^*;\boldsymbol{z})\|\mathbf{R}] \leq \frac{\varepsilon}{\gamma^2}\frac{1}{\mathbb{P}(\mathbf{R})}$ | $*$ | $\mathbb{E}_{\boldsymbol{z}}[\ell(\widehat{f};\boldsymbol{z}) - \ell(f^*;\boldsymbol{z})\|\mathbf{R}] \leq \frac{\varepsilon}{\gamma c}\frac{1}{\mathbb{P}(\mathbf{R})}$ | $\circ$ |
| large-variance diminishing $(1 - \mathbb{P}(\mathbf{R}))c \leq \varepsilon$ | $\mathbb{E}_{\boldsymbol{z}}[\ell(\widehat{f};\boldsymbol{z}) - \ell(f^*;\boldsymbol{z})] \leq \frac{\varepsilon}{\gamma^2}$ | $*$ | $\mathbb{E}_{\boldsymbol{z}}[\ell(\widehat{f};\boldsymbol{z}) - \ell(f^*;\boldsymbol{z})] \leq \frac{\varepsilon}{\gamma c}$ | $\circ$ |

Table 1: Comparison between existing results and ours. Let $\boldsymbol{z} \triangleq (\boldsymbol{x}, y)$, $\diamond$ represents results that are implied by Massart & Nédélec (2006), where the excessive risk bound is minimax optimal, $*$ represents results that are implied by Bartlett et al. (2005), $\circ$ represents results in this work. Throughout the table, all results are derived given sample size $\tilde{\Theta}(\frac{1}{\varepsilon})$. In classification task, $f^*(\boldsymbol{x})$ is the Bayes optimal classifier and $\omega^*(\boldsymbol{x})$ is the margin function calculated as $2\mathbb{P}[y = f^*(\boldsymbol{x})\|\boldsymbol{x}] - 1$. $\widehat{\omega}(\boldsymbol{x})$ is the approximated margin and $\gamma$ represents the minimum margin. For the regression task, $f^*(\boldsymbol{x}) = \mathbb{E}[y\|\boldsymbol{x}]$, $\sigma^{2*}(\boldsymbol{x}) = \mathbf{Var}(y\|\boldsymbol{x})$ and $\widehat{\sigma^2}(\boldsymbol{x})$ is an approximation of $\sigma^{2*}(\boldsymbol{x})$. $1/\gamma$ is taken to be the maximum possible variance. We denote $\mathbb{P}(\cdot)$ as the probability mass of a set of events. For the general region, weighted ERM recovers at least the same rate as ERM. For the large-margin region or low-variance region, weighted ERM improves ERM by $(c/\gamma) > 1$ on the conditional excessive risk bound. In the case of classification we provide a lower bound construction showing the tightness of the conditional risk (Theorem 4.4). The improved conditional excessive risk of weighted-ERM also implies an improved original excessive risk under low-margin or large-variance diminishing condition. To clarify, it is not necessary for the value of $c$ to be 'large.' Instead, having $c$ as a constant, such as 0.1, is sufficient, especially when $\gamma$ approaches a vanishing value, e.g., when $\gamma$ is of order $\sqrt{\varepsilon}$.

## 2 Related Work

In this section, we provide a brief literature review of related work. On the theory front, we review some representative results in standard ERM, under both the slow rate and the fast rate regimes; on the methodology front, we establish connections to existing literature where Gaussian maximum likelihood estimation (MLE) is performed in heteroscedastic regression settings, and note that MLE coincides with weighted ERM when $\ell_2$ loss is used with the inverse variance function being the weight. Finally, given the connection between the results established in this work and those in selective learning, we also briefly review results for the case where a rejection option is allowed.

**Empirical risk minimization**  The combination of VC-tool and Rademacher complexity analysis (Vapnik & Chervonenkis, 1974; Koltchinskii & Panchenko, 2000; Mendelson, 2002b; Boucheron et al., 2005; Bartlett et al., 2005) constitutes one of the most widely-adopted frameworks in deriving risk bounds for ERM. Under the "slow rate" regime, its generalization error bound is at the rate of $O(n^{-1/2})$, and such rate can be established using a uniform convergence argument (Vapnik & Chervonenkis, 1974; Talagrand, 1994; Boucheron et al., 2005) over an "unrestricted" function class. On the other hand, fast rate can be achieved if one moves away from an "arbitrary" hypothesis (Mendelson, 2002a; Boucheron et al., 2005; Bartlett et al., 2005). By restricting to a subset of the function class that has small variance and considering the Rademacher averages associated with this small subset (i.e., "localized") as a complexity term, sharper bounds can be obtained vis-à-vis the case where global averages are used. In particular, under a Bernstein-type condition (Massart & Nédélec, 2006), fast rate up to $O(n^{-1})$ can be obtained in well-specified (or realizable, equivalently) settings where the optimal model lies within the hypothesis class. For binary classification, this condition can be satisfied by imposing Massart or Tsybakov noise condition (Mammen & Tsybakov, 1999; Tsybakov, 2004; Massart & Nédélec, 2006). The same fast rate can be obtained for bounded regression (Bartlett et al., 2005; Massart & Nédélec, 2006) or learning problems whose loss function satisfies strong convexity and Lipschitz condition (Klochkov & Zhivotovskiy, 2021), or when it is self-concordant (Bach, 2010) or exp-concave (Koren & Levy, 2015).

**Maximum likelihood estimation and weighted ERM**  The weighted empirical risk minimization has been adopted to tackle several challenges in machine learning including distribution shifting (Cortes et al., 2008; 2010; Ge et al., 2024), censored observation (Ausset et al., 2022) and reinforcement learning (Jiang & Li, 2016; Xie et al., 2019; Min et al., 2021). In particular, in heteroscedastic regression settings where the variance depends on the input, a negative Gaussian likelihood-based formulation coincides with weighted ERM, when the latter uses $\ell_2$ loss and the inverse of the conditional variance as the weight. In particular, in the view of weighted ERM, samples with higher conditional variance (and potentially more noisy) are down-weighted and therefore contribute less to the overall empirical risk. The conditional variance can be estimated using either parametric models (Daye et al., 2012; Zhang et al., 2023) or kernel methods (Cawley et al., 2004). Despite the wide applicability of such a formulation (Kendall & Gal, 2017; Lakshminarayanan et al., 2017; Shah et al., 2022; Seitzer et al., 2022), little has been done in comparing the sample efficiency between estimators based on the weighted and those of the standard ERM. In this work, we aim to fill this gap, by explicitly analyzing the sample efficiency in the weighted ERM case and juxtaposing its performance to that in the standard ERM.

**Reject option and selective risk**  Along a line of work that slightly deviates from the ERM is the learning with rejection option; e.g., Chow's reject option model (Chow, 1970). The model allows one to refrain from making predictions on "hard" instances at the inference stage with some abstention cost; better precision is obtained in exchange for lower coverage, and the selective risk is only evaluated on the covered subset. Such a learning schema has applications in domains such as finance (Pidan & El-Yaniv, 2011) and health care (Hanczar & Dougherty, 2008), and can be generalized to other tasks (Mozannar & Sontag, 2020). Extensive work has been done to understand the trade-off between coverage ratio and precision (Herbei & Wegkamp, 2006; Bartlett & Wegkamp, 2008; Yuan & Wegkamp, 2010; El-Yaniv et al., 2010; Cortes et al., 2016), along with the statistical properties (Bousquet & Zhivotovskiy, 2021; Zhang et al., 2024) associated with the learning procedure. At the conceptual level, weighted ERM can be viewed as a "soft" counterpart to learning with the rejection option. In particular, instead of adopting a hard "in-or-out" rule to improve selective risk, performing weighted ERM can lead to similar benefit in a soft manner, with the weight typically coming from plug-in estimates (Herbei & Wegkamp, 2006; Bartlett & Wegkamp, 2008; Franc et al., 2023). In practice, the weight can be the estimated margin or the inverse variance function, in classification and regression settings, respectively.

## 3  A Road Map

**Notation**  We denote vectors by bold-faced letters (e.g., $\boldsymbol{x}$) and scalar variables by lower-case regular ones (e.g., $y$). Let $(\mathcal{X}, \mathbb{P}, \mathcal{B})$ be a probability space, where $\mathcal{X}$ denotes the sample space, $\mathbb{P}$ the probability measure and $\mathcal{B}$ the Borel $\sigma$-field. The expectation with respect to the probability distribution of a random

vector $\boldsymbol{x} \in \mathcal{X}$ is denoted by $\mathbb{E}_{\boldsymbol{x}}[\cdot]$; the subscript is omitted whenever there is no ambiguity in the respective context. For a random vector $\boldsymbol{z} \triangleq (\boldsymbol{x}, y)$ defined on the product sample space $\mathcal{Z} \triangleq \mathcal{X} \times \mathcal{Y}$ (equipped with the corresponding Borel $\sigma$-field), its probability measure is denoted by $\bar{\mathbb{P}}$. We use $\boldsymbol{z} \sim \mathcal{D}$ to denote that $\boldsymbol{z}$ is sampled from data generative process (DGP) $\mathcal{D}$. Throughout the remainder of this manuscript, we use $\boldsymbol{x}_i$ with a subscript $i$ to denote the $i$th random sample drawn from $\mathcal{X}$; $y_i$ and $\boldsymbol{z}_i$ are analogously defined. We also denote $\mathcal{W} : \mathcal{X} \mapsto [0, c_1]$ as a hypothesis classes with finite peudo-dimension $d_P < \infty$. Finally, we use $\tilde{\Theta}(\cdot)$ and $\tilde{O}(\cdot)$ to denote counterparts of the standard $\Theta(\cdot), O(\cdot)$ notation while suppressing the poly-logarithmic factors; the notation $f \gtrsim g$ means $f \geq cg$ for some universal constant $c$, and $\lesssim$ is analogously defined. We say $g \asymp f$ if $\underline{c}g \leq f \leq \bar{c}g$ for some universal constants $\bar{c}$ and $\underline{c}$.

**Problem setup**  Consider a sequence of i.i.d. samples $\boldsymbol{z}_{1:n} \triangleq \{\boldsymbol{z}_i = (\boldsymbol{x}_i, y_i)\}_{i=1}^n$ drawn from sample space $\mathcal{Z}$, with $\boldsymbol{x}_i$'s being the input and $y_i$'s the output; let $f : \mathcal{X} \mapsto \mathcal{Y}, f \in \mathcal{F}$ and $\ell$ be some loss function that is bounded and Lipschitz. In this study, we focus on analyzing the excess risk of the ERM and the weighted ERM estimators (see definitions below) on certain sub-regions.

**Definition 1** (Empirical risk and the ERM estimator). Given $\boldsymbol{z}_{1:n}$ and $f \in \mathcal{F}$, for loss function $\ell : \mathcal{F} \times \mathcal{Z} \to [0, a]$, we define the empirical loss as

$$L_{\mathrm{ERM}}(f; \boldsymbol{z}_{1:n}) = \tfrac{1}{n} \sum\nolimits_{i=1}^n \ell(f; \boldsymbol{z}_i);$$

the ERM estimator is given by

$$\widehat{f}_{\mathrm{ERM}} \triangleq \underset{f \in \mathcal{F}}{\arg\min}\, L_{\mathrm{ERM}} = \underset{f \in \mathcal{F}}{\arg\min}\, \tfrac{1}{n} \sum\nolimits_{i=1}^n \ell(f; \boldsymbol{z}_i). \tag{3}$$

**Definition 2** (Weighted empirical risk and the weighted ERM estimator). Given $\boldsymbol{z}_{1:n}$, $f \in \mathcal{F}$ and some weight function $\omega \in \mathcal{W} : \mathcal{X} \mapsto [0, c_1]$, for loss function $\ell : \mathcal{F} \times \mathcal{Z} \mapsto [0, a]$, we define the weighted empirical loss as

$$L_{\mathrm{wERM}}(f; \boldsymbol{z}_{1:n}) = \tfrac{1}{n} \sum\nolimits_{i=1}^n \omega(\boldsymbol{x}_i)\ell(f; \boldsymbol{z}_i);$$

the weighted ERM estimator is correspondingly given in the form of

$$\widehat{f}_{\mathrm{wERM}} \triangleq \underset{f \in \mathcal{F}}{\arg\min}\, L_{\mathrm{wERM}} = \underset{f \in \mathcal{F}}{\arg\min}\, \tfrac{1}{n} \sum\nolimits_{i=1}^n \omega(\boldsymbol{x}_i)\ell(f; \boldsymbol{z}_i). \tag{4}$$

We provide a brief overview of the main results established in Section 4. As briefly mentioned in Section 1, under the weighted ERM schema, an improved bound can be derived by encapsulating the weight function inside the appropriate terms of the Bernstein-type condition, namely,

$$\underbrace{\mathbf{Var}[h] \leq B\mathbb{E}[h]}_{\text{(vanilla Bernstein-type condition)}} \qquad \to \qquad \underbrace{\mathbf{Var}[\omega(\boldsymbol{x})h(\boldsymbol{x})] \leq \mathbb{E}[\omega(\boldsymbol{x})h(\boldsymbol{x})]}_{\text{(balanceble Bernstein condition)}}.$$

Such an "encapsulation" step does not alter the rate of $O(n^{-1})$ in the risk bound; however, the constant on which the bound depends will change, which then leads to a tighter bound for selected regions, as manifested through an improved *problem-dependent* constant. This selected region is determined by the ratio $B/(B'\omega(\boldsymbol{x}))$; both $B$ and $B'$ potentially depend on some $\gamma$ that characterizes the uniform lower or upper bound of the weight, depending on the setting (classification or regression). Finally, note that a Bernstein-type condition can be satisfied by imposing some margin condition or boundedness, in classification setting with 0-1 loss and heteroscedastic regression settings with $\ell_2$ loss, respectively.

To derive the desired risk bounds, we adopt the following road map and formalize the arguments in Section 4.

- We first establish the risk bounds of the weighted ERM estimator (in Theorem 4.1 for classification and in Theorem 4.5 for regression, resp.), assuming that the weight function $\widehat{\omega}(\boldsymbol{x})$ used in the estimation is sufficiently close to "true" one $\omega^*(\boldsymbol{x})$. Note that $\omega(\cdot)$ coincides with the margin function in the case of classification and the inverse of the variance function in the case of regression.

- Subsequently in Theorem 4.3 (for classification) and Theorem 4.6 (for regression), we provide the risk bound for $\widehat{\omega}(\boldsymbol{x})$, which shows that the sample complexity for estimating $\omega(\boldsymbol{x})$ is comparable to that of learning the $f^*(\cdot)$. This justifies the aforementioned assumption on $\widehat{\omega}$, in that such an assumption can be operationalized without rendering the estimation "harder".

- Finally, for the classification case, we additionally show that the established conditional excess risk bounds for the weighted ERM estimator is tight, by providing a lower bound result that matches its *conditional* (i.e., in sub-regions) excess risk upper bound. To contrast, the ERM estimator is provably inferior to the weighted one in terms of the lower bound in such sub-regions, despite of its minimax optimality in the *entire* region.

  Note that the above result points to the fact that the weight function (i.e., the margin function) we consider is optimal, as manifested by a minimax optimal conditional excessive risk bound by adopting such a weight function.

Note that our results point to the possibility of leveraging weighted ERM to achieve superior performance in certain regions, provided that the weight function is carefully designed and the region is chosen accordingly. See also results presented in Section 5 based on synthetic data experiments that attests to our theoretical claims, as well as how an estimate of the weight function can be readily obtained from data, and the weighted ERM schema be operationalized in two steps.

## 4 Main Results

Before stating our main results that are applicable to a more general setting, we first present results for two specific ones, namely, classification under margin condition (Section 4.1) and bounded heteroskedastic regression (Section 4.2) settings. Our result suggests that weighted ERM can improve the standard ERM error bound by a problem-dependent constant, in regions with high margin in the former and those with low variance in the latter. In Section 4.3, we present our main results for a more general setting; specifically, we show that a similar property for weighted ERM holds, as long as the loss function under consideration satisfies a "balanceable" Bernstein type condition.

### 4.1 Classification with/without margin condition

We formalize the classification problem considered in this paper, which largely follows from the general setting of Massart/benign label noise (Massart & Nédélec, 2006; Hanneke, 2009; Diakonikolas et al., 2019).

Let $\mathcal{F} \triangleq \{f : \mathcal{X} \mapsto \{-1, 1\}\}$ be a hypothesis class with finite VC-dimension[*] $d_{\mathrm{VC}}(\mathcal{F}) < \infty$, $\mathcal{G} \triangleq \{\eta : \mathcal{X} \mapsto [0, 1]\}$ be a hypothesis class with finite pseudo-dimension[†] $d_P(\mathcal{G}) < \infty$, and $\mathcal{W} \triangleq \{\omega : \mathcal{X} \mapsto [\gamma, 1]\}$ be some other hypothesis class with some fixed constant $0 \leq \gamma \leq 1$.

The data generative process (DGP) for $\boldsymbol{z} \triangleq (\boldsymbol{x}, y)$ can be characterized as follows:

$$\boldsymbol{x} \in \mathcal{X}; \qquad y|\boldsymbol{x} = \begin{cases} 1 & \text{with probability } \eta^*(\boldsymbol{x}) \\ -1 & \text{with probability } 1 - \eta^*(\boldsymbol{x}) \end{cases}, \quad \text{where} \quad \eta^*(\boldsymbol{x}) \triangleq \mathbb{P}(y = 1|\boldsymbol{x}). \tag{5}$$

For the family of DGP described in (5), $\eta^*(\boldsymbol{x})$ captures the conditional probability. The label $y$ can be equivalently viewed as satisfying $y|\boldsymbol{x} \sim 2\mathrm{Ber}(\eta^*(\boldsymbol{x})) - 1$, where $\mathrm{Ber}(p)$ denotes a Bernoulli random variable with success rate $p$. Let $f^*$ be the target hypothesis or the Bayes optimal classifier, defined as

$$f^*(\boldsymbol{x}) \triangleq \arg\max_{c \in \{-1, 1\}} \mathbb{P}(y = c|\boldsymbol{x}),$$

---

[*]The VC dimension of $\mathcal{F} = \{f : \mathcal{X} \mapsto \{-1, 1\}\}$ is the largest integer $d$ such that $S_{\mathcal{F}}(d) = 2$, where $S_{\mathcal{F}}(k)$ is the value of the growth function, namely, the largest cardinality $\{(f(x_1), f(x_2), ..., f(x_k)) : f \in \mathcal{F}\}$ among all $x_1, ..., x_k \in \mathcal{X}$ (Vapnik & Chervonenkis, 1971).

[†]The pseudo-dimension of $\mathcal{G} = \{g : \mathcal{X} \mapsto [l, u]\}$ is the VC-dimension of the hypothesis class $\mathcal{H} = \{h : \mathcal{X} \times \mathbb{R} \mapsto \{-1, 1\} \mid h(x, t) = \mathrm{sign}(g(x) - t), g \in \mathcal{G}, t \in \mathbb{R}, x \in \mathcal{X}\}$ (Pollard, 1990).

that is, $f^*(\boldsymbol{x})$ is the value of $c \in \{-1, 1\}$ that maximizes the conditional probability, which also satisfies $f^*(\boldsymbol{x}) = 2\mathbb{1}\{\eta^*(\boldsymbol{x}) \geq \frac{1}{2}\} - 1$. Let $\omega^*$ be the feature-dependent *margin function* given by $\omega^*(\boldsymbol{x}) \triangleq 2\,\mathbb{P}(y = f^*(\boldsymbol{x})|\boldsymbol{x}) - 1$, and note that $\omega^*(\boldsymbol{x}) \equiv |\eta^*(\boldsymbol{x}) - 1/2|$. Throughout the remainder of this subsection, we consider the well-specified setting (Massart & Nédélec, 2006; Bousquet & Zhivotovskiy, 2021) by assuming $f^* \in \mathcal{F}$, $\eta^* \in \mathcal{G}$ and $\omega^* \in \mathcal{W}$.

We choose $\ell(f, \boldsymbol{z}) = \mathbb{1}\{f(\boldsymbol{x}) \neq y\}$ as the loss function and propose to use the margin function $\omega^*(\boldsymbol{x})$ as the weight to perform a weighted ERM; this leads to a weighted risk of the form $\omega^*(\boldsymbol{x})\mathbb{1}\{f(\boldsymbol{x}) \neq y\}$. In practice, when $\omega^*(\boldsymbol{x})$ is not available, one can use its estimate while still achieving similar results under mild assumptions.

We first give an informal overview of the results established. Typically, a standard margin condition (Massart & Nédélec, 2006; Bousquet & Zhivotovskiy, 2021) requires the minimum margin to satisfy $\gamma \triangleq \inf_{\boldsymbol{x}} \omega^*(\boldsymbol{x}) > 0$. For standard ERM, Bernstein-type condition (Bartlett et al., 2005) is satisfied in the form of:

$$\mathbf{Var}_{\boldsymbol{z}}[\mathbb{1}\{y \neq f(\boldsymbol{x})\} - \mathbb{1}\{y \neq f^*(\boldsymbol{x})\}] \leq (1/\gamma) \cdot \mathbb{E}_{\boldsymbol{z}}[(\mathbb{1}\{y \neq f(\boldsymbol{x})\} - \mathbb{1}\{y \neq f^*(\boldsymbol{x})\})]$$

whereas in the case of weighted ERM, it is satisfied in the following form:

$$\mathbf{Var}_{\boldsymbol{z}}[\omega^*(\boldsymbol{x})(\mathbb{1}\{y \neq f(\boldsymbol{x})\} - \mathbb{1}\{y \neq f^*(\boldsymbol{x})\})] \leq \mathbb{E}_{\boldsymbol{z}}[\omega^*(\boldsymbol{x})(\mathbb{1}\{y \neq f(\boldsymbol{x})\} - \mathbb{1}\{y \neq f^*(\boldsymbol{x})\})]. \tag{6}$$

Crucially, in the latter case, the $1/\gamma$ factor is removed, which subsequently leads to an improved bound. In particular, Equation (6) does not require the margin condition $\gamma > 0$ (Massart & Nédélec, 2006), i.e., $\gamma$ can be zero; this suggests that even if the vanilla empirical risk does not satisfy the Bernstein condition, it is still possible to utilize a weight function to establish a Bernstein-type condition. Theorem 4.1 formally states this result.

**Theorem 4.1** (Risk Bound for the case of Classification). *Suppose that we have $\widehat{\omega}(\cdot) \in \mathcal{W}$ s.t. $\mathbb{E}_{\boldsymbol{x}}[(\widehat{\omega}(\boldsymbol{x}) - \omega^*(\boldsymbol{x}))^2] \leq \varepsilon$ is satisfied. Let $S_n = \{(\boldsymbol{x}_i, y_i)\}_{i=1}^n$ be i.i.d. samples drawn according to the DGP described in (5), and they are independent of the ones used for estimating $\widehat{\omega}$. Let $\widehat{f}_{wERM}$ be the weighted ERM estimator as defined in (4), with the weight substituted by $\widehat{\omega}(\boldsymbol{x}_i)$'s. Then the following hold simultaneously with probability at least $1 - \delta$ for some $\varepsilon > 0, \delta > 0$:*

$$\mathbb{E}_{\boldsymbol{z}}[\omega^*(\boldsymbol{x})(\ell(\widehat{f}_{wERM}; \boldsymbol{z}) - \ell(f^*; \boldsymbol{z}))] \leq \varepsilon, \qquad \mathbb{E}_{\boldsymbol{z}}[\widehat{\omega}(\boldsymbol{x})(\ell(\widehat{f}_{wERM}; \boldsymbol{z}) - \ell(f^*; \boldsymbol{z}))] \leq \varepsilon, \tag{7}$$

*provided that the sample size $n$ satisfies $n \gtrsim \frac{d_{VC}(\mathcal{F})\log(\frac{1}{\varepsilon}) + \log(\frac{1}{\delta})}{\varepsilon}$.*

The next theorem shows that there exists a DGP that conforms with the description in (5), such that when the estimation is performed based on samples drawn from it, the risk of the weighted ERM estimator on some sub-region can be arbitrarily close to zero, provided that the sample size grows commensurately; on the other hand, the risk of the ERM estimator on the same region is bounded below by a constant, irrespective of the sample size. Here the sub-region is characterized by a large-margin level set, and the risk on this sub-region can be viewed as the selective risk.

**Theorem 4.2.** *There exists a DGP that belongs to the DGP family satisfying (5), such that under the same assumption as in Theorem 4.1, when the estimation is performed based on i.i.d. samples $S_n = \{(\boldsymbol{x}_i, y_i)\}_{i=1}^n$ drawn from it, the following hold simultaneously for the ERM estimator (as per defined in (3)) and weighted ERM estimator (as per defined in (4) with the weight function substituted by $\widehat{\omega}(\boldsymbol{x}_i)$'s), with sample size $n = \frac{64d \log(d) \log(\frac{1}{\delta})}{\gamma\varepsilon}$:*

- *with probability at least $0.1$, $\mathbb{E}_{\boldsymbol{x}}[\mathbb{1}\{\widehat{f}_{ERM} \neq f^*\}|\omega^*(\boldsymbol{x}) > \gamma] \geq 0.015$;*
- *with probability at least $1 - \delta$, $\mathbb{E}_{\boldsymbol{x}}[\mathbb{1}\{\widehat{f}_{wERM} \neq f^*\}|\omega^*(\boldsymbol{x}) > \gamma] \lesssim \varepsilon$.*

*Remark* 1 (On the bounds established). Some discussion of the implications of the bounds established in Theorems 4.1 and 4.2 is provided next. First note that the low/high margin region can be characterized through $\{\boldsymbol{x} : \omega(\boldsymbol{x}) < c\}$ (and $\{\boldsymbol{x} : \omega(\boldsymbol{x}) > c\}$, resp.); the *excess risk bound*, given by $\mathbb{E}_{\boldsymbol{z}}[\mathbb{1}\{\widehat{f} \neq y\} - \mathbb{1}\{f^* \neq y\}]$, can be further derived for weighted ERM based on (7). This enables a direct comparison between the bound of the weighted ERM and that of ERM, depending on the property of $\mathbb{P}(\omega^*(\boldsymbol{x}) \leq c)$.

- **Improved bounds in the large-margin region.** For any $c \in [0, 1)$, given large-margin region characterized by $\{\boldsymbol{x} : \omega^*(\boldsymbol{x}) > c\}$ with $\mathbb{P}(\omega^*(\boldsymbol{x}) > c) > 0$ and $\tilde{\Theta}(\frac{1}{\varepsilon})$ samples, the risk bound of an ERM estimator (e.g., Massart & Nédélec, 2006, equation (7)) implies the following excess risk within the region:

$$\mathbb{E}_{\boldsymbol{z}}[\mathbb{1}\{\widehat{f}_{ERM}(\boldsymbol{x}) \neq y\} - \mathbb{1}\{f^*(\boldsymbol{x}) \neq y\} | \omega^*(\boldsymbol{x}) \geq c] \leq \frac{\varepsilon}{\gamma \mathbb{P}(\omega^*(\boldsymbol{x}) > c)};$$

  for the weighted ERM estimator, it achieves

$$\mathbb{E}_{\boldsymbol{z}}[\mathbb{1}\{\widehat{f}_{wERM}(\boldsymbol{x}) \neq y\} - \mathbb{1}\{f^*(\boldsymbol{x}) \neq y\} | \omega^*(\boldsymbol{x}) \geq c] \leq \frac{\varepsilon}{c \mathbb{P}(\omega^*(\boldsymbol{x}) > c)}, \tag{8}$$

  which improves the ERM bound by a factor of $(\gamma/c)$. In particular, in Theorem 4.2, a lower bound construction for the conditional risk on the large-margin region is presented, to demonstrate that there exists a scenario where the ERM estimator fails with constant probability, while the weighted ERM one achieves standard PAC guarantee (Valiant, 1984). Note that under Chow's rejection rule (Chow, 1970) which optimally balances the trade-off between coverage and accuracy, the non-reject region coincides precisely with the large-margin region described above.

  These analyses establish the benefit of the weighted ERM procedure where data is weighed by the margin function $\omega^*(\boldsymbol{x})$. In particular, its major advantage lies in the region where $c$ is large compared to $\gamma$, and such advantage vanishes as $(\gamma/c) \to 1$. Later in Theorem 4.4, a pathological scenario achieving a matching lower bound is presented, which implies the minimax optimality of the bound presented in (8).

- **Improved bounds under the "low-margin diminishing" condition.** The low-margin diminishing condition holds when there exists $c$ such that $\mathbb{P}(\omega^*(\boldsymbol{x}) \leq c)c^2 \leq \varepsilon$. One can view the set $\{\boldsymbol{x} | \omega^*(\boldsymbol{x}) \leq \sqrt{\varepsilon}\}$ as the collection of $\boldsymbol{x}$ with "low margin", whose corresponding $y$'s have high label noise. The condition $\mathbb{P}(\omega^*(\boldsymbol{x}) \leq c)c^2 \leq \varepsilon$ describes settings where the low margin set has diminishing mass. Under such a condition, Equation (8) implies the following improved excessive risk bound:

$$\mathbb{E}_{\boldsymbol{z}}[\mathbb{1}\{\widehat{f} \neq y\} - \mathbb{1}\{f^* \neq y\}] \leq \varepsilon/c, \tag{9}$$

  which enjoys improvement of a factor $\gamma/c$ when compared to the excessive risk of ERM. The condition $\mathbb{P}(\omega^*(\boldsymbol{x}) \leq c)c^2 \leq \varepsilon$ could be satisfied when $c \lesssim \sqrt{\varepsilon}$ and $\mathbb{P}(\omega^*(\boldsymbol{x}) \leq c) \lesssim 0.1$. The derivation is presented in Appendix A.2.

- **Fast rate with/without margin condition.** Under standard margin condition $\gamma > 0$, based on the Corollary 3 from Massart & Nédélec (2006), both ERM and weighted ERM achieves a $\mathbb{E}_{\boldsymbol{z}}[\omega^*(\boldsymbol{x})(\mathbb{1}\{\widehat{f} \neq y\} - \mathbb{1}\{f^* \neq y\})] \leq \varepsilon$ excessive risk with $\tilde{\Theta}(\frac{1}{\varepsilon})$ samples, which is the information-theoretic optimality (Massart & Nédélec, 2006; Zhivotovskiy & Hanneke, 2018). Notably, Theorem 4.1 implies such a result given the fact that $\omega^*(\boldsymbol{x}) \geq \gamma$.

  When the standard margin condition is not satisfied, e.g., for cases $\boldsymbol{x} \in \mathcal{X}$ with $\eta^*(\boldsymbol{x}) = 0.5$, the corresponding $y$ effectively becomes a pure noise. In this case, a fast rate of $O(1/n)$ cannot be attained due to these "extremely noisy" data points. However, it is still possible to attain a fast rate with a slight modification of the vanilla empirical risk. Specifically, in Theorem 2.1 of Bousquet & Zhivotovskiy (2021) establishes risk bounds that enjoy a fast rate under Chow's rejection option framework (Chow, 1970), without requiring a margin condition. This is achieved by introducing an "artificial" margin through the inclusion of a rejection option, which allows for the removal of those "extremely noisy data points" during both the learning and inference processes. Similarly, the weighted-ERM approach follows a similar principle by utilizing a weight function $\omega^*(\boldsymbol{x})$ to down-weigh such extremely noisy data points with $\omega^*(\boldsymbol{x}) = 0$.

Both Theorems 4.1 and 4.2 require that the surrogate $\widehat{\omega}$ be reasonably close to $\omega^*$, and such a condition is presented in the form of $\mathbb{E}_{\boldsymbol{x}}[(\widehat{\omega}(\boldsymbol{x}) - \omega^*(\boldsymbol{x}))^2] \leq \varepsilon$. One may wonder if the task of approximating $\omega^*$ is statistically more challenging than learning the classifier itself in terms of sample efficiency. Theorem 4.3 addresses this concern and suggests that under standard assumptions, the required condition holds with high probability and the sample complexity of learning $\omega^*(\cdot)$ is comparable to that of learning $f^*(\cdot)$.

**Theorem 4.3** (Risk bound for estimating $\omega^*$). *Given i.i.d. samples $S'_n = \{(\boldsymbol{x}_i, y_i)\}_{i=1}^n$ drawn from the DGP described in* (5)*, let $\widehat{\eta} = \arg\min_{\eta \in \mathcal{G}} \frac{1}{n} \sum_{\boldsymbol{z}_i \in S'_n} (\eta(\boldsymbol{x}_i) - y_i)^2$. Then the following holds with probability at least $1 - \delta$ for some $\varepsilon > 0, \delta > 0$:*

$$\mathbb{E}_{\boldsymbol{x}}[(\widehat{\eta}(\boldsymbol{x}) - \eta^*(\boldsymbol{x}))^2] \leq \varepsilon,$$

*provided that the sample size $n$ satisfies $n \gtrsim \frac{d_P(\mathcal{G}) \log(\frac{1}{\varepsilon}) + \log(\frac{1}{\delta})}{\varepsilon}$. Further, let $\widehat{\omega}(\boldsymbol{x}) \equiv |\widehat{\eta}(\boldsymbol{x}) - 1/2|$, and the above result further implies that*

$$\mathbb{E}_{\boldsymbol{x}}[(\widehat{\omega}(\boldsymbol{x}) - \omega^*(\boldsymbol{x}))^2)] \leq \varepsilon.$$

*Remark* 2. Both Theorems 4.1 and 4.2 effectively assume the availability of a sufficiently-well estimated weight function $\widehat{\omega}(\cdot)$, and that the subsequent weighted estimation procedure is carried out on samples $S_n$ independent of those used for obtaining $\widehat{\omega}(\cdot)$. In practice, this can be operationalized via sample-splitting; namely, one equally splits the training set into two halves at random, and uses one half for weight estimation and the other half for obtaining $\widehat{f}$. Such a procedure would increase the generalization error bound by a factor of two, in exchange for the problem-dependent constant improvement in the large margin sub-region.

The following corollary can be derived by combining Theorems 4.1 and 4.3, which takes into account all the randomness embedded in the training samples:

**Corollary 1.** *Under the assumptions in Theorems 4.1 and 4.3, let $\widehat{\omega}$ be the one defined in Theorem 4.3, then the following hold simultaneously with probability at least $1 - 2\delta$ for some $\varepsilon > 0$:*

$$\mathbb{E}_{\boldsymbol{z}}[\omega^*(\boldsymbol{x})(\ell(\widehat{f}_{wERM}; \boldsymbol{z}) - \ell(f^*; \boldsymbol{z}))] \leq \varepsilon, \qquad \mathbb{E}_{\boldsymbol{z}}[\widehat{\omega}(\boldsymbol{x})(\ell(\widehat{f}_{wERM}; \boldsymbol{z}) - \ell(f^*; \boldsymbol{z}))] \leq \varepsilon. \tag{10}$$

Next we present our lower bound results for the conditional excessive risk bound.

**Theorem 4.4.** *There exists a set of DGPs that is in accordance with the description in* (5)*, such that for i.i.d. samples $S_n = \{(\boldsymbol{x}_i, y_i)\}_i^n$ drawn from (any) such DGPs with $n \asymp \frac{1}{\varepsilon}$, the following inequality holds:*

$$\inf_{f \in \mathcal{F}} \sup_{\mathcal{D}} c \mathbb{E}_{\boldsymbol{z} \sim \mathcal{D}}[\mathbb{1}\{f(\boldsymbol{x}) \neq y\} - \mathbb{1}\{f^*(\boldsymbol{x}) \neq y\}, \omega^*(\boldsymbol{x}) \geq c] \geq \varepsilon.$$

Theorem 4.4 suggests that for all $f \in \mathcal{F}$, there exists $\mathcal{D}$ where the following holds:

$$\mathbb{E}_{\boldsymbol{z} \sim \mathcal{D}}[\mathbb{1}\{f(\boldsymbol{x}) \neq y\} - \mathbb{1}\{f^*(\boldsymbol{x}) \neq y\}|\omega^*(\boldsymbol{x}) \geq c] \geq \frac{\varepsilon}{c\mathbb{P}(\omega^*(\boldsymbol{x}) > c)};$$

in other words, there exists a family of data generating process satisfying (5), such that the conditional excessive risk of *any* estimator—irrespective of the learning procedure—cannot improve upon the bound established in (8), in the absence of any additional assumptions. The theorem implies that the conditional excessive risk bound of weighted ERM estimator in (8) is tight.

**Selective inference in practice**   While the optimal weight function $\omega^*(\boldsymbol{x})$ is not available in practice, Theorem 4.1 admits an $\varepsilon$-approximation of $\omega^*(\boldsymbol{x})$ in the PAC sense, that is, one can have control over the selective risk of the weighted ERM estimator reasonably well using *any* "good" estimates of the margin function. Note that using the plug-in estimate is a standard procedure adopted in the literature related to selective classification (Herbei & Wegkamp, 2006), where users have the option to abstain from predicting data with high uncertainty or low margin. This is pertinent in scenarios where prioritizing accuracy in the low uncertainty region (conditional risk) takes precedence over accuracy across the entire domain (unconditional risk). The result in Theorem 4.1 suggests the same, that by using a good estimate of the margin to re-weight the empirical risk, one can improve the conditional risk at the inference stage.

## 4.2   Bounded heteroscedastic regression

The regression problem considered in this section is formalized next. Let $y \in \mathbb{R}$ be generated according to

$$y = f^*(\boldsymbol{x}) + \sqrt{\sigma^{2*}(\boldsymbol{x})} \cdot \xi, \quad \boldsymbol{x} \in \mathcal{X}, \tag{11}$$

where $f^*(\boldsymbol{x}) \triangleq \mathbb{E}(y|\boldsymbol{x})$, $\sigma^{2*}(\boldsymbol{x}) \triangleq \mathbf{Var}(Y|\boldsymbol{x})$, and $\xi \in (-c_2, c_2)$ is some bounded noise with zero mean and unit variance. $f^*(\boldsymbol{x})$ is effectively the target hypothesis.

Without loss of generality, let $\mathcal{F} \triangleq \{f : \mathcal{X} \mapsto [-1,1]\}$, $\mathcal{G} \triangleq \{\sigma^2 : \mathcal{X} \mapsto [c_3, 1/\gamma]\}, 0 < c_3 < 1$ be hypothesis classes with finite pseudo-dimensions $d_P(\mathcal{F}) < \infty$ and $d_P(\mathcal{G}) < \infty$, respectively. Note that for the range of $\sigma^2$, we assume that $c_3$ is bounded away from 0 and therefore the variance is non-vanishing; on the other hand, the upper bound satisfies $1/\gamma \geq 1$ and thus we do not preclude scenarios where the variance dominates the mean function, similar to the settings considered in Zhang et al. (2023). Throughout this subsection, we consider the well-specified learning setting, namely, $f^*(\boldsymbol{x}) \in \mathcal{F}, \sigma^{2*}(\boldsymbol{x}) \in \mathcal{G}$. Additionally, let $\mathcal{W} \triangleq \{\omega : \mathcal{X} \mapsto [\gamma, 1/c_3]\}$ be some other hypothesis class for the weight function $\omega^*(\boldsymbol{x})$, which satisfies $\omega^*(\boldsymbol{x}) \equiv 1/(\sigma^2)^*(\boldsymbol{x})$

To perform weighted ERM, we adopt the mean-squared-error loss given by $\ell(f, \boldsymbol{z}) = (y - f(\boldsymbol{x}))^2$, and a data-dependent weight that coincides with the inverse of the *variance function* $\sigma^{*2}(\boldsymbol{x})$; this leads to weighted loss function of the form $\frac{(y-f(\boldsymbol{x}))^2}{\sigma^{2*}(\boldsymbol{x})}$. Note that in practice, $\sigma^{2*}(\boldsymbol{x})$ is unavailable and one can replace it with some estimate, while still achieving similar results, provided that the estimate approximates the truth reasonably well.

We first give a high-level account of the results established. Let $\sigma^{2*}(\boldsymbol{x}) \leq 1/\gamma$ be the maximum variance as defined in Zhang et al. (2023), then the following holds (see derivation in the appendix):

$$\mathbf{Var}_{\boldsymbol{z}}[(y - f(\boldsymbol{x}))^2 - (y - f^*(\boldsymbol{x}))^2] \leq (8/\gamma) \cdot \mathbb{E}_{\boldsymbol{z}}[(y - f(\boldsymbol{x}))^2 - (y - f^*(\boldsymbol{x}))^2]. \tag{12}$$

The expression in (12) suggests that the Bernstein-type condition is satisfied with $B = 4/\gamma$; an analysis similar to Corollary 3.7 in Bartlett et al. (2005) further leads to the following risk bound $\mathbb{E}_{\boldsymbol{x}}[(f^*(\boldsymbol{x}) - \widehat{f}_{\mathrm{ERM}}(\boldsymbol{x}))^2] = O\left(\frac{1}{\gamma^2 n}\right)$. For weighted ERM, $\sigma^*(\boldsymbol{x})$ (or $\omega^*(\boldsymbol{x}) \equiv 1/\sigma^{2*}(\boldsymbol{x})$, equivalently) is introduced to "balance" the inequality in (12), resulting in the Bernstein-type condition to hold in the following form:

$$\mathbf{Var}_{\boldsymbol{z}}\left[\frac{C(y - f(\boldsymbol{x}))^2}{\sigma^{2*}(\boldsymbol{x})} - \frac{C(y - f^*(\boldsymbol{x}))^2}{\sigma^{2*}(\boldsymbol{x})}\right] \leq \mathbb{E}_{\boldsymbol{z}}\left[\frac{C(y - f(\boldsymbol{x}))^2}{\sigma^{2*}(\boldsymbol{x})} - \frac{C(y - f^*(\boldsymbol{x}))^2}{\sigma^{2*}(\boldsymbol{x})}\right], \tag{13}$$

where $C = 1/2(1 + 4/c_3)$. Once again, leveraging the results in Bartlett et al. (2005) gives $\mathbb{E}_{\boldsymbol{x}}[\frac{1}{\sigma^{2*}(\boldsymbol{x})}(f^*(\boldsymbol{x}) - \widehat{f}_{\mathrm{wERM}}(\boldsymbol{x}))^2] = O\left(\frac{1}{\gamma n}\right)$, which effectively removes the $1/\gamma$ factor. These statements are formally stated next.

**Theorem 4.5.** *Suppose we have $\widehat{\sigma}^2 \in \mathcal{G}$ s.t. $\mathbb{E}_{\boldsymbol{x}}[(1/\widehat{\sigma}^2(\boldsymbol{x}) - 1/\sigma^{2*}(\boldsymbol{x}))^2] \leq \varepsilon/c_3^2$. Given i.i.d samples $S_n = \{(\boldsymbol{x}_i, y_i)\}_{i=1}^n$ that are drawn according to the DGP and independent of those used for obtaining $\widehat{\sigma}^2$, let $\widehat{f}_{wERM}$ be the weighted ERM estimator defined in Equation (4) with the weight function substituted by $1/\widehat{\sigma}^2(\cdot)$; then the following holds simultaneously with probability at least $1 - \delta$ for some $\varepsilon > 0, \delta > 0$:*

$$\mathbb{E}_{\boldsymbol{x}}\left[\frac{1}{\sigma^{2*}(\boldsymbol{x})}(\widehat{f}_{wERM}(\boldsymbol{x}) - f^*(\boldsymbol{x}))^2\right] \leq \varepsilon \quad and \quad \mathbb{E}_{\boldsymbol{x}}\left[\frac{1}{\widehat{\sigma}^2(\boldsymbol{x})}(\widehat{f}_{wERM}(\boldsymbol{x}) - f^*(\boldsymbol{x}))^2\right] \leq \varepsilon,$$

*provided that the sample size $n$ satisfies $n \gtrsim \frac{d(\mathcal{F})(\log(\frac{1}{\varepsilon}) + \log(1/\gamma) - \log(c_3) + \log(\frac{1}{\delta}))}{\gamma \varepsilon c_3^2}$.*

*Remark* 3. The risk bounds in Theorem 4.5 could be made analogous to those in Theorem 4.1. A standard analysis using techniques in Bartlett et al. (2005) implies that the ERM estimator achieves $\mathbb{E}_{\boldsymbol{x}}\left[(\widehat{f}_{\mathrm{ERM}}(\boldsymbol{x}) - f^*(\boldsymbol{x}))^2\right] \leq \varepsilon$ using $\tilde{\Theta}\left(\frac{1}{\gamma^2 \varepsilon}\right)$ samples whereas the weighted ERM one achieves $\mathbb{E}_{\boldsymbol{x}}\left[\frac{1}{\sigma^{2*}(\boldsymbol{x})}(\widehat{f}_{\mathrm{wERM}}(\boldsymbol{x}) - f^*(\boldsymbol{x}))^2\right] \leq \varepsilon$ with $\tilde{\Theta}\left(\frac{1}{\gamma \varepsilon}\right)$ samples. Similar to the conclusions in the classification task, weighted ERM achieves sample efficiency at least comparable to ERM in the general region, and is superior in the small-variance region as depicted by $\sigma^{2*}(\boldsymbol{x}) \leq 1/c$, with a problem-dependent constant $\gamma/c$ improvement. Additionally, by using the negative log-likelihood loss, the sample complexity of learning $\sigma^{2*}(\cdot)$ is comparable to that of learning $f^*(\cdot)$, as presented next in Theorem 4.6.

Next we provide some guarantees for learning the $\sigma^2(\boldsymbol{x})$ function. Here we seek to obtain $\widehat{\sigma}^2$ by minimizing the negative log-likelihood loss, while restricting $(\mu, \sigma^2)$ to be in the hypothesis class $\widetilde{\mathcal{F}} \times \widetilde{\mathcal{G}}, \widetilde{\mathcal{F}} \subseteq \mathcal{F}, \widetilde{\mathcal{G}} \subseteq \mathcal{G}$, so that the normalized residual square is bounded: $\frac{(f^*(\boldsymbol{x}) - \mu(\boldsymbol{x}) + \rho \cdot \sqrt{\sigma^{2*}(\boldsymbol{x})})^2}{\sigma^2(\boldsymbol{x})} \leq 4c_2^2, \rho \in (-c_2, c_2), \forall \boldsymbol{x} \in \mathcal{X}$.

**Theorem 4.6** (Risk bound for estimating $\sigma^{2*}$)**.** *Let $S'_n = \{(\boldsymbol{x}_i, y_i)\}_{i=1}^n$ be i.i.d. samples drawn according to the DGP in* (11) *and*

$$\left(\widehat{\mu}, \widehat{\sigma}^2\right) \triangleq \arg\min_{(\mu, \sigma^2) \in \widetilde{\mathcal{F}} \times \widetilde{\mathcal{G}}} \frac{1}{n} \sum_{(\boldsymbol{x}_i, y_i) \in S'_n} \left[ \log(\sigma^2(\boldsymbol{x}_i)) + \frac{(y_i - \mu(\boldsymbol{x}_i))^2}{\sigma^2(\boldsymbol{x}_i)} \right],$$

*Then for any $\varepsilon > 0, \delta > 0$, the following holds with probability at least $1 - \delta$:*

$$\mathbb{E}_{\boldsymbol{x}} \left[ \left( \frac{1}{\widehat{\sigma}^2(\boldsymbol{x})} - \frac{1}{\sigma^{2*}(\boldsymbol{x})} \right)^2 \right] \le \varepsilon,$$

*provided that the sample size satisfies*

$$n \gtrsim \frac{\mathcal{T}_1 \mathcal{T}_2^3 \left( (d_P(\mathcal{G}) + d_P(\mathcal{F})(\mathcal{T}_3 + \mathcal{T}_4 + \mathcal{T}_5) \right)}{c_3^2 \varepsilon},$$

*where $\mathcal{T}_1 = (1 + c_2^2 + 1/c_3^2)$, $\mathcal{T}_2 = (c_2^2 + \log^2(1/\gamma))$, $\mathcal{T}_3 = \log(1/c_3^2 + c_2^2/c_3^2 + c_2^2/c_3^2\gamma)$, $\mathcal{T}_4 = \log(\frac{1}{\varepsilon})$, $\mathcal{T}_5 = \log(\frac{1}{\delta})$.*

Note that the PAC guarantee for learning $\widehat{\sigma}^2(\boldsymbol{x})$ has been studied in Zhang et al. (2023) which admits a bound at the rate of $\tilde{O}\big(\frac{1}{\sqrt{n}}\big)$, whereas the bound in Theorem 4.6 is of the order $\tilde{O}\big(\frac{1}{n}\big)$. See Appendix A.9 for a discussion that compares the two and the proof for the above theorem.

The following corollary combines Theorems 4.5 and 4.6 and takes into account all the randomness embedded in the samples:

**Corollary 2.** *Under the setting considered in Theorem 4.5 and Theorem 4.6, with $\widehat{\sigma}^2(\boldsymbol{x})$ be the one defined in Theorem 4.6, the following hold simultaneously with probability at least $1 - 2\delta$:*

$$\mathbb{E}_{\boldsymbol{x}} \left[ \frac{1}{\sigma^{2*}(\boldsymbol{x})} (\widehat{f}(\boldsymbol{x}) - f^*(\boldsymbol{x}))^2 \right] \le \varepsilon \quad and \quad \mathbb{E}_{\boldsymbol{x}} \left[ \frac{1}{\widehat{\sigma}^2(\boldsymbol{x})} (\widehat{f}(\boldsymbol{x}) - f^*(\boldsymbol{x}))^2 \right] \le \varepsilon.$$

### 4.3 General case

Next, we generalize the classification and regression examples presented in Sections 4.1 and 4.2 above. Consider hypothesis class $\mathcal{F} \triangleq \{f : \mathcal{X} \mapsto [-1, 1]\}$ with complexity measure $d(\mathcal{F}) < \infty$, and $\mathcal{W} \triangleq \{\omega : \mathcal{X} \mapsto [\gamma, c_1]\}$ with complexity measure $d(\mathcal{W}) < \infty$, and $c_1 > 0, \gamma > 0^{\ddagger}$. Assume that the target hypothesis $f^* \in \mathcal{F}$ and the true weight $\omega^* \in \mathcal{W}$.

A *balanceable Bernstein-type condition* is given in the following assumption, together with some other technical assumptions required for the loss function.

**Assumption 1.** *Let $\mathcal{D} : \mathcal{F} \times \mathcal{F} \times \mathcal{X} \mapsto [0, b]$ be uniformly bounded function that captures the excess risk. The following are assumed to hold for the loss function $\ell(\cdot, \cdot)$:*

- **Lipschitzness and uniform boundedness.**

$$\begin{aligned}
\forall \boldsymbol{z}, f \quad &|\ell(f, \boldsymbol{z}) - \ell(f^*, \boldsymbol{z})| \le a; \\
\forall \boldsymbol{z}, f_1, f_2 \quad &|\ell(f_1, \boldsymbol{z}) - \ell(f_2, \boldsymbol{z})| \le L|f_1 - f_2|.
\end{aligned} \tag{14}$$

- **Under semi-random noise label.** Suppose the DGP satisfies semi-random noise label (Diakonikolas et al., 2021; Pia et al., 2022) and the following are satisfied:

$$\begin{aligned}
\mathbb{E}_{\boldsymbol{z}}[\ell(f, \boldsymbol{z}) - \ell(f^*, \boldsymbol{z})] &= \mathbb{E}_{\boldsymbol{x}}[\omega^*(\boldsymbol{x}) \mathcal{D}(f^*(\boldsymbol{x}), f(\boldsymbol{x}))] \\
\mathbb{E}_y[(\ell(f, \boldsymbol{z}) - \ell(f^*, \boldsymbol{z}))^2] &\le \mathcal{D}(f^*(\boldsymbol{x}), f(\boldsymbol{x}))
\end{aligned} \tag{15}$$

---

$^{\ddagger}$Note that here we effectively assume that $\omega^*$ is bounded away from zero to accommodate both the case of classification and regression. However, note that in the case of classification, the assumption $\omega^*$ being bounded away from zero is equivalent to the margin condition from Massart & Nédélec (2006). The weighted ERM framework is actually agnostic to this condition and allows $\omega^*$ to be zero.

- **Balanceable Bernstein condition.** Under the same semi-random noise label assumption for the DGP, there exists a uniformly bounded $\omega(\boldsymbol{x})$ that the following holds:

$$\mathbf{Var}_{\boldsymbol{z}}[\omega(\boldsymbol{x})(\ell(f,\boldsymbol{z}) - \ell(f^*,\boldsymbol{z}))] \leq \mathbb{E}_{\boldsymbol{z}}[\omega(\boldsymbol{x})(\ell(f,\boldsymbol{z}) - \ell(f^*,\boldsymbol{z}))]. \tag{16}$$

For (16), if one replaces $\omega^*$ by its uniform lower bound, standard Bernstein-type condition $\mathbf{Var}[h(\boldsymbol{x})] \leq B\mathbb{E}_{\boldsymbol{x}}[h(\boldsymbol{x})]$ can be recovered with $\omega(\boldsymbol{x}) = 1$ and $B = 1/\gamma$. On the other hand, there exists $\omega(\boldsymbol{x})$ that balances the ratio between $\mathbf{Var}[\omega(\boldsymbol{x})h(\boldsymbol{x})]$ and $\mathbb{E}[\omega(\boldsymbol{x})h(\boldsymbol{x})]$, such that Bernstein-type condition holds with an improved multiplier. In particular, one can set $\omega(\cdot) \triangleq \omega^*(\cdot)$ so that $B = 1$, as in (6) and (13).

*Remark* 4 (Assumption 1 in classification and regression settings). We provide concrete examples for how expressions in Assumption 1 manifest in classification and regression settings, respectively.

- For classification settings, let $\ell(f;\boldsymbol{z}) \triangleq \mathbb{1}\{f(\boldsymbol{x}) \neq y\}$ and $\mathcal{D}(f^*, \widehat{f}, \boldsymbol{x}) \triangleq \mathbb{1}\{f^*(\boldsymbol{x}) \neq \widehat{f}(\boldsymbol{x})\}$. It is easy to verify that (14) is satisfied with $L = 1, a = 1$; (15) is satisfied with $\omega^*(\boldsymbol{x}) \triangleq |2\eta^*(\boldsymbol{x}) - 1|$ (see derivation in Appendix A.1); and (16) holds as established in (6)

- For regression settings, let $\ell(f;\boldsymbol{z}) \triangleq (y - f(\boldsymbol{x}))^2$ and $\mathcal{D}(f^*, \widehat{f}, \boldsymbol{x}) \triangleq \frac{\sigma^{2*}(\boldsymbol{x})}{C}(f^*(\boldsymbol{x}) - \widehat{f}(\boldsymbol{x}))^2$ for some constant $C$. (14) is satisfied with $a = 8 + (8c_2^2/\sqrt{\gamma}), L = c_2/\sqrt{\gamma}$ where $b = 4/(C\gamma), c_1 = \frac{1}{2}$; (15) holds with $\omega^*(\boldsymbol{x}) \triangleq \frac{C}{\sigma^{2*}(\boldsymbol{x})}$; and (16) holds as established in (13).

**Theorem 4.7.** *Suppose Assumption 1 holds. Let $f^* \in \mathcal{F}$ and $\omega^* \in \mathcal{W}$, and suppose we have $\widehat{\omega} \in \mathcal{W}$ s.t. $\mathbb{E}_{\boldsymbol{x}}[(\widehat{\omega}(\boldsymbol{x}) - \omega^*(\boldsymbol{x}))^2] \leq \frac{\varepsilon}{b}$. Given i.i.d. samples $S_n = \{(\boldsymbol{x}_i, y_i)\}_{i=1}^n$ drawn from the data generative process, let $\widehat{f}_{wERM} \triangleq \arg\min_{f \in \mathcal{F}} \frac{1}{n} \sum_{\boldsymbol{z}_i \in S_n} \widehat{\omega}(\boldsymbol{x})\ell(f;\boldsymbol{z}_i)$. Then for any $\varepsilon > 0, \delta > 0$, the following holds simultaneously with probability at least $1 - \delta$:*

$$\mathbb{E}_{\boldsymbol{x}}[\widehat{\omega}^2(\boldsymbol{x})\mathcal{D}(f^*, \widehat{f}_{wERM}, \boldsymbol{x})] \leq \varepsilon, \quad \mathbb{E}_{\boldsymbol{x}}[\omega^{*2}(\boldsymbol{x})\mathcal{D}(f^*, \widehat{f}_{wERM}, \boldsymbol{x})] \leq \varepsilon, \text{ and } \quad \mathbb{E}_{\boldsymbol{x}}[\widehat{\omega}(\boldsymbol{x})\omega^*(\boldsymbol{x})\mathcal{D}(f^*, \widehat{f}_{wERM}, \boldsymbol{x})] \leq \varepsilon,$$

*as long as the sample size requirement is satisfied:*

$$n \gtrsim \frac{c_1^2 a^2(d(\mathcal{F})\log(\frac{1}{\varepsilon}) + \log(c_1 L) + \log(\frac{1}{\delta}))}{\varepsilon} + \frac{c_1 a \log(\frac{1}{\delta})}{\varepsilon}.$$

*Remark* 5. The proof of Theorem 4.7 is deferred to the appendix. A major hurdle in completing the proof comes from the inaccessibility of $\omega^*(\boldsymbol{x})$ and thus one needs to use $\widehat{\omega}(\boldsymbol{x})$ instead; this is out of practical consideration as one can only access an estimated version. To overcome this challenge, it suffices to require $\mathbb{E}_{\boldsymbol{x}}[(\widehat{\omega}(\boldsymbol{x}) - \omega^*(\boldsymbol{x}))^2] \leq \varepsilon/b$, with which one can show the weighted empirical risk satisfies an $\varepsilon$ additive error version of the Bernstein type condition, namely, $\mathbf{Var}[h] \leq B\mathbb{E}[h] + \varepsilon$. To this end, we prove an alternative version of Theorem 3.3 in Bartlett et al. (2005) under this relaxed Bernstein-type condition, and show that the weighted ERM achieves a fast rate in the generalization error bound.

## 5 Synthetic Data Experiments

We present results from synthetic data experiments to support our theoretical claims, respectively for regression and classification settings. For both settings, we follow a two-step procedure, in which we first perform ERM to obtain estimates for the mean and the weight, followed by a reweighting step. Subsequently, we compare two sets of estimates—resp. from ERM and weighted ERM—in terms of their selective risk, where the selective set is chosen over a range with varying coverage determined by the variance or the margin, depending on the setting under consideration. For both experiments, the size of the training set is set at 2e4, to ensure that the algorithm has access to adequate number of samples and circumvent any potential issues due to lack of fit, although empirically the conclusion broadly holds even with much smaller sample sizes.

### 5.1 Experiments under regression settings

We consider regression settings in the presence of heteroscedasticity, similar to the ones used in Skafte et al. (2019); Seitzer et al. (2022). The true data generating process is given by a univariate regression with $x \in \mathbb{R}$ of the form

$$y = f^*(x) + \sqrt{\sigma^{2*}(x)} \cdot \xi, \quad \mathbb{E}(\xi) = 0; \ \mathbf{Var}(\xi) = 1.$$

The mean $f^*(x) \triangleq x\sin(x)$ is a sinusoidal function; the scale function of the additive noise $\xi$ depends on the value of $x$, and is given by $(\sigma^2)^*(x) \triangleq (0.09)(1 + x^2)$. The regressor $x$ is sampled uniformly from $[0, 10]$ and the noise $\xi$ is standard Gaussian. Figure 1a provides a visualization of the data resulting from this DGP.

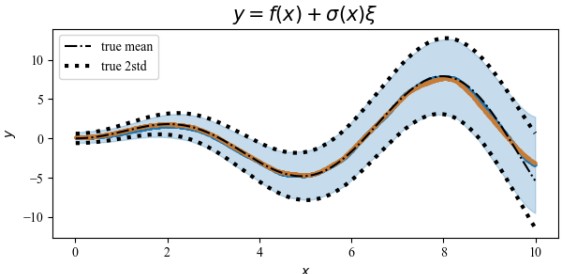

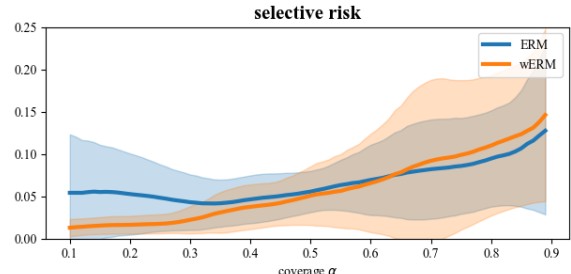

(a) Visualization of the DGP and the estimates. The true mean $f^*(x)$ and the two-standard deviation bands are in black dotted lines. The mean estimates from ERM (blue) and wERM (orange) are in solid lines, and the shaded area corresponds to the two-standard deviation derived from the variance estimate of the ERM step.

(b) Risk over the selective set with varying coverage $\alpha$; the solid lines correspond the risk on the test (sub)set averaged over 10 runs of the experiment, and the shaded area correspond to 1 standard deviation.

Figure 1: Regression setting: underlying true data, estimates from ERM and weighted ERM, and the selective risk

We consider the following estimation procedure using $\ell_2$ loss; $f(x)$ and $\sigma^2(x)$ are both parametrized by multi-layer perceptrons (MLP):

- An ERM step that gives rise to the mean and the variance estimates, that is,

$$\widehat{f}_{\mathrm{ERM}}(x) := \arg\min_f \sum_{i=1}^n (y_i - f(x_i))^2 \quad \text{and} \quad \widehat{\sigma}^2(x) := \arg\min_{\sigma^2} \sum_{i=1}^n \Big[ \log \sigma^2(x) + \frac{(y_i - \widehat{f}_{\mathrm{ERM}}(x_i))^2}{\sigma^2(x)} \Big];$$

- A reweighting step, with the weight given by the precision estimate from the ERM step:

$$\widehat{f}_{\mathrm{wERM}}(x) := \arg\min_f \sum_{i=1}^n \omega(x_i)\big(y_i - f(x_i)\big)^2 \quad \text{where} \quad \omega(x_i) := 1/\widehat{\sigma}^2(x_i).$$

Once $\widehat{f}_{\mathrm{ERM}}(x)$ and $\widehat{f}_{\mathrm{wERM}}(x)$ are obtained, on the test set, we consider evaluating their risk over a range of selective set with varying coverage $\alpha \in [0, 1]$. Concretely, at evaluation time, the selective risk is calculated as

$$\mathcal{R}_\alpha := \mathbb{E}\Big[\big(f^\star(x) - f(x)\big)^2 \mid \big\{\sigma^2(\boldsymbol{x}) \leq q_\alpha(\sigma^2)\big\}\Big],$$

where $q_\alpha(\sigma^2)$ is the $\alpha$ quantile of the variance over the entire domain; a low-coverage (i.e., small $\alpha$) selective set corresponds to the low-variance region. Empirically, the risk is obtained by substituting $f$ by either $\widehat{f}_{\mathrm{ERM}}$ or $\widehat{f}_{\mathrm{wERM}}$ for each test data point in the selective set then taking the average, with $\widehat{\sigma}(x)$ coming from the ERM step. The cut-off $q_\alpha(\sigma^2)$ is determined by the empirical quantile of the estimated $\sigma^2$ on the validation set.

Figure 1a presents the mean estimate from ERM (blue) and weighted ERM (orange) respectively, although the quality of the fit is satisfactory for both cases and therefore they largely overlap with the truth and becomes hard to distinguish, visually. Figure 1b displays the risk over the selective set with varying coverage $\alpha$. As it can be seen from the plot, weighted ERM has an advantage over the ERM estimate in the low-coverage region, as manifested by a lower selective risk, and the advantage diminishes as the coverage $\alpha$ increases. This is in accordance with the theoretical results established in Section 4.2.

As a remark, the practical implication for such results is that in certain scenarios (e.g., some finance applications) where one takes actions only when there is high confidence and abstains otherwise (and therefore being "selective"), a weighted ERM procedure can be leveraged to obtain more refined estimates for the region of interest where actions would take place.

## 5.2 Experiments under classification settings

For classification, we consider the following data-generating process for illustration purpose, in which extremely noisy data points are present. The features $\boldsymbol{x}_i \in \mathbb{R}^2$ are sampled from class-conditional Gaussian with equal covariance, that is $\mathbb{P}(c_i = k) = p_k$; $\mathbb{P}(\boldsymbol{x}_i|c_i = k) \sim \mathcal{N}(\boldsymbol{\mu}_k, \Sigma)$. Here we consider 4 "clusters" with $k \in \{0, 0', 1, 1'\}$, where $p_{0'} = 0.5, p_0 = 0.25, p_1 = 0.20, p_{1'} = 0.05$; $\boldsymbol{\mu}_{0'} = (-10, 0)^\top, \boldsymbol{\mu}_0 = (-3, 0)^\top, \boldsymbol{\mu}_1 = (3, 0)^\top, \boldsymbol{\mu}_{1'} = (12, 0)^\top, \Sigma = \begin{bmatrix} 2 & 0.5 \\ 0.5 & 2 \end{bmatrix}$. Let

$$\phi^*(\boldsymbol{x}_i) \triangleq \mathbb{P}\big(c_i \in \{1, 1'\} \mid \boldsymbol{x}_i\big) = \big(\sum\nolimits_{k \in \{1,1'\}} p_k \cdot p(\boldsymbol{x}_i; \boldsymbol{\mu}_k, \Sigma)\big) / \big(\sum\nolimits_{k \in \{0,0',1,1'\}} p_k \cdot p(\boldsymbol{x}_i; \boldsymbol{\mu}_k, \Sigma)\big),$$

where $p(\boldsymbol{x}_i; \boldsymbol{\mu}_k, \Sigma)$ denotes the pdf corresponding to $\mathcal{N}(\boldsymbol{\mu}_k, \Sigma)$ evaluated at $\boldsymbol{x}_i$, and the equality follows from Bayes rule. Set $y_i^{\text{initial}} \triangleq \mathbb{1}\{\phi^*(\boldsymbol{x}_i) > 1/2\}$. Further, we inject noise to the class labels for points that are in cluster $0'$, such that their final labels are flipped with probability $p_{\text{flip}}$, that is:

$$y_i = 1 - y_i^{\text{initial}} \ \ (\text{w.p. } p_{\text{flip}}) \ \text{ if } c_i = 0' \quad \text{otherwise} \quad y_i^{\text{initial}}.$$

It is worth noting that given that above-mentioned DGP, the theoretical decision boundary is linear; additionally, the theoretical margin is given by $\omega^*(\boldsymbol{x}_i) \triangleq |\eta^*(\boldsymbol{x}_i) - 1/2|$, where $\eta^*(\boldsymbol{x}_i) \triangleq \mathbb{1}\{\phi^*(\boldsymbol{x}_i) > 1/2\}$ if $c_i \neq 0'$ otherwise $p_{\text{flip}}$. In this setting, the flipping probability $p_{\text{flip}}$ is set at 0.49.

Similar to the case of regression, we consider the following procedure that entails two steps, using the cross-entropy loss:

- An ERM step: we obtain the estimated margin $\widehat{\omega}(\boldsymbol{x}_i) = |\widehat{\eta}(\boldsymbol{x}_i) - 1/2|$ where $\widehat{\eta}(\boldsymbol{x}_i)$ is the estimated Bayes-optimal classifier, and a linear decision boundary that gives rise to the class labels $\widehat{f}(\boldsymbol{x}_i)_{\text{ERM}}$.

- A reweighting step with the weight set at $\widehat{w}(\boldsymbol{x}_i)$, which then gives rise to an updated decision boundary and the associated class label $\widehat{f}(\boldsymbol{x}_i)_{\text{wERM}}$.

The selective risk is then evaluated as

$$\mathcal{R}_\alpha := \mathbb{E}\Big[\mathbb{1}\{f^* \neq \widehat{f}\} \mid \big\{\omega(\boldsymbol{x}) \geq q_\alpha(\omega)\big\}\Big], \qquad f^* := \mathbb{1}\{\eta^* > 0.5\}.$$

where $q_\alpha(\omega)$ is the $\alpha$ quantile of the margin over the entire domain; a low-coverage (i.e., small $\alpha$) selective set corresponds to the large-margin region. Empirically, the evaluation is done by averaging the risk on the corresponding test (sub)set, and $y$ is substituted by either $\widehat{f}(\boldsymbol{x}_i)_{\text{ERM}}$ or $\widehat{f}(\boldsymbol{x}_i)_{\text{wERM}}$; the cut off $q_\alpha(\omega)$ is empirically determined by the quantile of the estimated margin on the validation set.

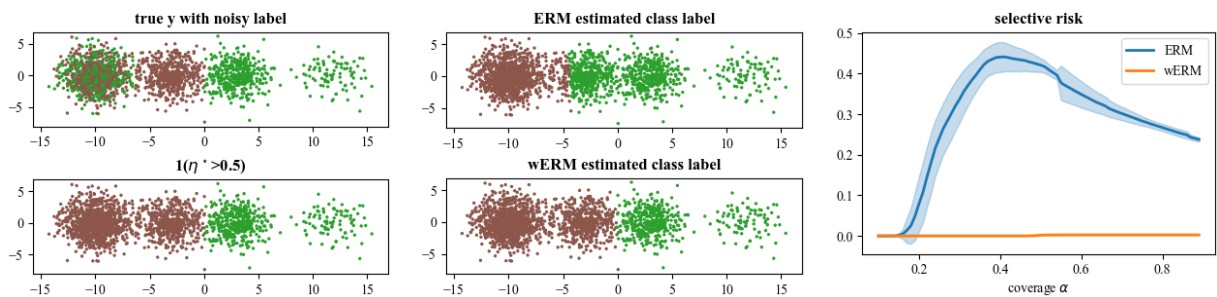

(a) Visualization of the DGP. The top panel displays the class labels after injecting label noise, and the bottom shows class labels based on the Bayes optimal classifier, respectively

(b) Estimated class labels based on ERM (top) and weighted ERM (bottom), respectively

(c) Risk over the select set with varying coverage $\alpha$. the solid lines correspond the risk on the test (sub)set averaged over 10 runs of the experiment, and the shaded area corresponds to 1 standard deviation.

Figure 2: Classification setting: underlying true data, estimates from ERM and weighted ERM and the selective risk

Figure 2a provides a representative view of the data resulting from this DGP; Figure 2b displays the estimated class labels from the ERM (top) and the weighted ERM (bottom) step, respectively. As it can be seen from the figure, the latter largely aligns with the class labels dictated by the Bayes-optimal classifier, whereas the former has a noticeable number of points near the decision boundary being mis-labeled. Figure 2c provides a comparison of the selective risk for the two sets of estimates. In this specific setting, weighted ERM dominates.

## 6 Discussion, Limitation, and Future Work

To conclude, this work investigates the generalization error bound of weighted ERM under the fast-rate regime. We show that by additionally incorporating a carefully-designed weight function in each loss term, estimators based on weighted ERM can achieve a tighter bound in selected regions by a problem-dependent constant, when compared with the one from standard ERM. This finding leads to practical applications where one can use plug-in estimates of the weight function to obtain superior performance in sub-regions through a two-step procedure, as demonstrated in our synthetic data experiments.

It is worth noting that recent work Zhai et al. (2023) considers a generalized reweighting scheme where samples are reweighted dynamically during training; in spirit, this is similar to the procedure considered in our work, yet the two differ in the following aspects: (1) the starting point of Zhai et al. (2023) is the generalization of distributional robust optimization (DRO) algorithms; in particular, under a DRO setting where distributional shift is present yet the test distribution is "close" to the training one, at the conceptual level, training should focus on the "hard" cases. In our work, on the contrary, the setting under consideration can be made analogous to soft abstention, and thus the weight schema further upweights the "easy" cases. (2) The result established in Zhai et al. (2023) states that the generalized reweighting procedure leads to a solution that is close to the ERM one, in that the points they converge to are close; this is largely done by analyzing the properties of the estimates between successive iterates. On the other hand, this work is concerned with the statistical error bound of the weighted ERM estimator—in particular, its superiority over the standard ERM one in the high-confidence region.

There are several limitations of this work and directions that could be further explored. Throughout the paper, we consider well-specified settings. There are several difficulties in regards to the extension to mis-specified settings where the target hypothesis can potentially live outside of the hypothesis class in question. Possible extension includes exploring mis-specified settings with surrogate losses using tools developed in Awasthi et al. (2022); Mao et al. (2023), and leveraging several recent results which show that fast rate could be achieved under mis-specified setting via model selection aggregation (Tsybakov, 2003; Bousquet & Zhivotovskiy, 2021; Kanade et al., 2022). Other recent work that touched upon this issue includes Puchkin & Zhivotovskiy (2022), where to establish the desired results the authors require both the diameter of the hypothesis class and the star number to be finite. Alternatively, tools that can work in specific setups may be introduced to characterize the approximation error; e.g., if one considers a setting similar to that in Kohler & Langer (2021), namely, the hypothesis class being a set of fully connected DNNs and the target hypothesis being in the class of $(p, C)$-smooth functions, then the approximation error bound can be calibrated. The results developed in this work can potentially be extended to these settings, which however requires more involved analysis to handle various components (e.g., the approximation error of the weight function $\widehat{\omega}(\boldsymbol{x})$) and very specific assumptions on the exact setup.

Separately, our analysis is limited to Bernstein-type condition (Lee et al., 1996; Bartlett et al., 2005) of the form $\mathbf{Var}[h] \leq B\mathbb{E}[h]$. To study classification problems under Tsybakov noise condition (Mammen & Tsybakov, 1999; Tsybakov, 2004) and regression with $\ell_p$ risk (Bartlett & Mendelson, 2006), a more generalized form $\mathbb{E}[h^2] \leq B(\mathbb{E}[h])^{\beta}, \beta \in (0, 1]$ is required. Finally, for some other settings such as Offset Rademacher Complexity (Liang et al., 2015) and "small-ball condition" (Mendelson, 2018), where the fast rate has been established, we are optimistic they can also enjoy some problem-dependent constant improvement by exploring the structure of semi-random noise label with a properly designed weight function.

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

# A  Proofs and Discussions

In this section, we include proofs for the results established in Section 4 and the corresponding discussions. We introduce additional notation and definitions that are used in the ensuing development.

Let $[n] \triangleq \{1, \cdots, n\}$. We use $\{\boldsymbol{x}\}_{i \in [n]} \triangleq \boldsymbol{x}_{1:n}$ to denote samples of $\boldsymbol{x}$ of size $n$; $\{\boldsymbol{z}\}_{i \in [n]} \equiv \boldsymbol{z}_{1:n} = \{(\boldsymbol{x}_i, y_i)\}_{i=1}^n$ is analogously defined, and they are i.i.d. samples drawn from $\bar{\mathbb{P}}$. Given $\{\boldsymbol{z}\}_{i \in [n]}$, we use $\bar{\mathbb{P}}_n$ to denote the empirical measure. We use $\|\cdot\|_p$ to denote the $\ell_p$ norm of vectors and $\|\cdot\|_{L_p}$ to denote the $L_p$ norm of random variables under $\mathbb{P}$. The indicator function $\mathbb{1}\{\boldsymbol{x} \in A\}$ equals 1 when the condition $x \in A$ is true and 0 otherwise. Denote $a \vee b = \max(a, b)$ and $a \wedge b = \min(a, b)$. Let $\sigma_1, ..., \sigma_n$ be $n$ independent *Rademacher* random variables. For a function $h : \mathcal{Z} \to \mathbb{R}$, define:

$$\bar{\mathbb{P}}_n h \triangleq \frac{1}{n} \sum_{i=1}^n h(\boldsymbol{z}_i), \bar{\mathbb{P}} h \triangleq \mathbb{E}_{\boldsymbol{z} \sim \bar{\mathbb{P}}} h(\boldsymbol{z}).$$

For a family of functions $\mathcal{H} \triangleq \{h : \mathcal{Z} \to \mathbb{R}\}$ the *Rademacher Complexity* and *Rademacher Average* is defined as:

$$\mathfrak{R}_n \mathcal{H} = \mathbb{E}_{\sigma_{1:n}} \left[ \sup_{h \in \mathcal{H}} \frac{1}{n} \sum_{i=1}^n \sigma_i h(\boldsymbol{z}_i) \right], \qquad \mathfrak{R} \mathcal{H} = \mathbb{E}_{\boldsymbol{z}_{1:n}, \sigma_{1:n}} \left[ \sup_{h \in \mathcal{H}} \frac{1}{n} \sum_{i=1}^n \sigma_i h(\boldsymbol{z}_i) \right]$$

We further define the *Local Rademacher Complexity* and *Local Rademacher Average* with radius $r$ as $\mathfrak{R}_n\{h \in \mathcal{H}, \mathbb{P}_n h^2 \leq r\}$ and $\mathfrak{R}\{h \in \mathcal{H}, \mathbb{P} h^2 \leq r\}$.

**Definition 3** (Star Hull; see also Bousquet et al. (2003); Bartlett et al. (2005))**.** The star hull of set of functions $\mathcal{F}$ is defined as

$$*\mathcal{F} \equiv \{\alpha f : f \in \mathcal{F}, \alpha \in [0, 1]\}$$

**Definition 4** (Sub-Root Function (Bousquet et al., 2003; Bartlett et al., 2005))**.** A function $\psi : \mathbb{R} \to \mathbb{R}$ is sub-root if

- $\psi$ is non-decreasing

- $\psi$ is non-negative

- $\psi(r)/\sqrt{r}$ is non-increasing

And we say $r^*$ is a fixed point of $\psi$ if $\psi(r^*) = r^*$.

The following definition for VC-class could be found in (Vapnik & Chervonenkis, 1971; Van Der Vaart et al., 1996). We include them here for the sake of completeness.

**Definition 5** (VC-dimension; Vapnik & Chervonenkis (1971))**.** The VC-dimension $d_{VC}(\mathcal{F})$ of a hypothesis class $\mathcal{F} = \{f : \mathcal{X} \mapsto \{1, -1\}\}$ is the largest cardinality of any set $S \subseteq \mathcal{X}$ such that $\forall \bar{S} \subseteq S, \exists f \in \mathcal{F}$:

$$f(\boldsymbol{x}) = \begin{cases} 1 & \text{if } \boldsymbol{x} \in \bar{S} \\ -1 & \text{if } \boldsymbol{x} \in S \setminus \bar{S} \end{cases}$$

**Definition 6** (Pseudo-dimension; Pollard (1990))**.** The Pseudo-dimension $d_P(\mathcal{F})$ of a real-valued hypothesis class $\mathcal{G} = \{g : \mathcal{X} \mapsto [l, u]\}$ is the VC-dimension of the hypothesis class

$$\mathcal{H} = \{h : \mathcal{X} \times \mathbb{R} \mapsto \{-1, 1\} \mid h(\boldsymbol{x}, t) = \text{sign}(g(\boldsymbol{x}) - t), g \in \mathcal{G}, t \in \mathbb{R}\}.$$

### A.1 Derivation of Equation 6

The following derivation establishes the connection between $\mathbb{E}_{\boldsymbol{z}}[\ell(f;\boldsymbol{z}) - \ell(f^*;\boldsymbol{z})]$ and $\mathbb{E}_{\boldsymbol{x}}[\omega^*(\boldsymbol{x})\mathbb{1}\{f \neq f^*\}]$, which could be found in Boucheron et al. (2005). We include here for completeness.

$$
\begin{aligned}
&\mathbb{E}_{\boldsymbol{z}}[\ell(f;\boldsymbol{z}) - \ell(f^*;\boldsymbol{z})] \\
=&\mathbb{E}_{\boldsymbol{x},y}[\mathbb{1}\{y \neq f(\boldsymbol{x})\} - \mathbb{1}\{y \neq f^*(\boldsymbol{x})\}] \\
=&\mathbb{E}_{\boldsymbol{x},y}[\mathbb{P}[y = f^*(x)]\mathbb{1}\{f^*(\boldsymbol{x}) \neq f(\boldsymbol{x})\} + \mathbb{P}[y \neq f^*(x)]\mathbb{1}\{f^*(\boldsymbol{x}) = f(\boldsymbol{x})\} - \mathbb{P}[y \neq f^*(\boldsymbol{x})]] \\
=&\mathbb{E}_{\boldsymbol{x},y}[\mathbb{P}[y = f^*(x)]\mathbb{1}\{f^*(\boldsymbol{x}) \neq f(\boldsymbol{x})\} - \mathbb{P}[y \neq f^*(x)]\mathbb{1}\{f^*(\boldsymbol{x}) \neq f(\boldsymbol{x})\}] \\
=&\mathbb{E}_{\boldsymbol{x},y}[(\mathbb{P}[y = f^*(x)] - \mathbb{P}[y \neq f^*(x)])\mathbb{1}\{f^*(\boldsymbol{x}) \neq f(\boldsymbol{x})\}\}] \\
=&\mathbb{E}_{\boldsymbol{x}}[\omega^*(\boldsymbol{x})\mathbb{1}\{f \neq f^*\}];
\end{aligned}
\tag{17}
$$

Next, we derive Equation 6:

$$
\begin{aligned}
&\mathbf{Var}_{\boldsymbol{x},y}[\omega^*(\boldsymbol{x})(\mathbb{1}\{y \neq f(\boldsymbol{x})\} - \mathbb{1}\{y \neq f^*(\boldsymbol{x})\})] \\
\leq&\mathbb{E}_{\boldsymbol{x},y}[\omega^{*2}(\boldsymbol{x})(\mathbb{1}\{y \neq f(\boldsymbol{x})\} - \mathbb{1}\{y \neq f^*(\boldsymbol{x})\})^2] \\
=&\mathbb{E}_{\boldsymbol{x}}[\omega^{*2}(\boldsymbol{x})\mathbb{1}\{f^*(\boldsymbol{x}) \neq f(\boldsymbol{x})\}] \\
=&\mathbb{E}_{\boldsymbol{x},y}[\omega^*(\boldsymbol{x})(\mathbb{1}\{y \neq f(\boldsymbol{x})\} - \mathbb{1}\{y \neq f^*(\boldsymbol{x})\})\}]
\end{aligned}
$$

### A.2 Derivation of Equation 9

To see Equation 9, we first bound $\mathbb{E}_{\boldsymbol{x}}[\mathbb{1}\{\omega^*(\boldsymbol{x}) \geq c\}\omega^*(\boldsymbol{x})\mathbb{1}\{f \neq f^*\}]$. By leveraging Equation (17) and $\mathbb{E}_{\boldsymbol{x},y}[\omega^*(\boldsymbol{x})(\mathbb{1}\{f \neq y\} - \mathbb{1}\{f^* \neq y\})] \leq \varepsilon$ the follow inequality holds:

$$
\mathbb{E}_{\boldsymbol{x},y}[\omega^*(\boldsymbol{x})(\mathbb{1}\{f \neq y\} - \mathbb{1}\{f^* \neq y\})] = \mathbb{E}_{\boldsymbol{x}}[\omega^{*2}(\boldsymbol{x})\mathbb{1}\{f \neq f^*\}] \leq \varepsilon
\tag{18}
$$

which implies the following inequality:

$$
\mathbb{E}_{\boldsymbol{x}}[\mathbb{1}\{\omega^*(\boldsymbol{x}) \geq c\}c\omega^*(\boldsymbol{x})\mathbb{1}\{f \neq f^*\}] \leq \mathbb{E}_{\boldsymbol{x}}[\mathbb{1}\{\omega^*(\boldsymbol{x}) \geq c\}\omega^{*2}(\boldsymbol{x})\mathbb{1}\{f \neq f^*\}] \leq \varepsilon
\tag{19}
$$

thus $\mathbb{E}_{\boldsymbol{x}}[\mathbb{1}\{\omega^*(\boldsymbol{x}) \geq c\}\omega^*(\boldsymbol{x})\mathbb{1}\{f \neq f^*\}] \leq \frac{\varepsilon}{c}$. In addition,

$$
\begin{aligned}
&\mathbb{E}_{\boldsymbol{x}}[\mathbb{1}\{\omega^*(\boldsymbol{x}) < c\}\omega^*(\boldsymbol{x})\mathbb{1}\{f \neq f^*\}] \\
=&\mathbb{E}_{\boldsymbol{x}}[\mathbb{1}\{\omega^*(\boldsymbol{x}) < c\}\omega^*(\boldsymbol{x})\mathbb{1}\{f \neq f^*\}|\omega^*(\boldsymbol{x}) < c]\mathbb{P}(\omega^*(\boldsymbol{x}) < c) \\
+&\mathbb{E}_{\boldsymbol{x}}[\mathbb{1}\{\omega^*(\boldsymbol{x}) < c\}\omega^*(\boldsymbol{x})\mathbb{1}\{f \neq f^*\}|\omega^*(\boldsymbol{x}) \geq c]\mathbb{P}(\omega^*(\boldsymbol{x}) \geq c) \\
=&\mathbb{E}_{\boldsymbol{x}}[\omega^*(\boldsymbol{x})\mathbb{1}\{f \neq f^*\}|\omega^*(\boldsymbol{x}) < c]\mathbb{P}(\omega^*(\boldsymbol{x}) < c) \\
\leq&c \cdot \mathbb{P}(\omega^*(\boldsymbol{x}) < c)
\end{aligned}
\tag{20}
$$

Equation (19) and Equation (20) combined gives

$$
\begin{aligned}
&\mathbb{E}_{\boldsymbol{x},y}[\mathbb{1}\{f \neq y\} - \mathbb{1}\{f^* \neq y\}] = \mathbb{E}_{\boldsymbol{x}}[\omega^*(\boldsymbol{x})\mathbb{1}\{f \neq f^*\}] \\
=&\mathbb{E}_{\boldsymbol{x}}[\mathbb{1}\{\omega^*(\boldsymbol{x}) \geq c\}\omega^*(\boldsymbol{x})\mathbb{1}\{f \neq f^*\}] + \mathbb{E}_{\boldsymbol{x}}[\mathbb{1}\{\omega^*(\boldsymbol{x}) < c\}\omega^*(\boldsymbol{x})\mathbb{1}\{f \neq f^*\}] \\
\leq&\mathbb{P}(\omega^*(\boldsymbol{x}) < c)c + \frac{\varepsilon}{c}.
\end{aligned}
$$

### A.3 Derivation of Equation 12

Equation 12 is a standard result, we include the derivation here for the sake of completeness.

$$
\begin{aligned}
\mathbf{Var}_{\boldsymbol{x},y}[(y - f(\boldsymbol{x}))^2 - (y - f^*(\boldsymbol{x}))^2] &= \mathbf{Var}_{\boldsymbol{x},y}[(f^*(\boldsymbol{x}) - f(\boldsymbol{x}))(f^*(\boldsymbol{x}) + f(\boldsymbol{x}) - 2f^*(\boldsymbol{x}) - 2\xi\sqrt{\sigma^{2*}(\boldsymbol{x})})] \\
&\leq \mathbb{E}_{\boldsymbol{x},\xi}[(f^*(\boldsymbol{x}) - f(\boldsymbol{x}))^2(f(\boldsymbol{x}) - f^*(\boldsymbol{x}) - 2\xi\sqrt{\sigma^{2*}(\boldsymbol{x})})^2] \\
&\leq \mathbb{E}_{\boldsymbol{x}}[(f^*(\boldsymbol{x}) - f(\boldsymbol{x}))^2((f(\boldsymbol{x}) - f^*(\boldsymbol{x}))^2 + 4\sigma^{2*}(\boldsymbol{x}))] \\
&= \mathbb{E}_{\boldsymbol{x}}[\mathbb{E}_{y}[(f^*(\boldsymbol{x}) - y)^2 - (y - f(\boldsymbol{x}))^2]((f(\boldsymbol{x}) - f^*(\boldsymbol{x}))^2 + 4\sigma^{2*}(\boldsymbol{x}))] \\
&\leq \frac{8}{\gamma}\mathbb{E}_{\boldsymbol{x},y}[((f^*(\boldsymbol{x}) - y)^2 - (y - f(\boldsymbol{x}))^2)]
\end{aligned}
$$

### A.4 Proof of Theorem 4.1

The proof readily follows from invoking Theorem 4.7 with $\ell(f; \boldsymbol{z}) \triangleq \mathbf{1}\{f(\boldsymbol{x} \neq y)\}$, $\omega^*(\boldsymbol{x}) = |2\eta^*(\boldsymbol{x}) - 1|$, $\mathcal{D}(f_1, f_2, \boldsymbol{x}) = \mathbb{1}\{f_1 \neq f_2\}$. It can be easily verified that Assumption 1 is satisfied with $L = 1, a = 1, b = 1, \gamma = \inf_{\boldsymbol{x}} |2\eta^*(\boldsymbol{x}) - 1|$.

### A.5 Proof of Theorem 4.3

*Proof.* Let $R(\eta, \boldsymbol{z}_i) \triangleq (\eta(\boldsymbol{x}_i) - y_i)^2$. We define following function class:

$$\mathcal{H} \equiv \Delta \circ R \circ \mathcal{G} \equiv \left\{ \Delta R(\eta; \eta^*, \boldsymbol{z}) = R(\eta; \boldsymbol{z}) - R(\eta^*; \boldsymbol{z}) : \eta \in \mathcal{G} \right\}.$$

which is a composite function of hypothesis class $\mathcal{G}$, loss function $R$ and the difference operation $\Delta$. For simplicity we let $\Delta_{R,\eta} = \Delta R(\eta; \eta^*, \boldsymbol{z})$ when there is no ambiguity.

By definition of $\widehat{\eta}$, we have $\bar{\mathbb{P}}_n R(\widehat{\eta}; z) \leq \bar{\mathbb{P}}_n R(\eta^*; z)$, also by definition of $\Delta_{R,\widehat{\eta}}$ we have $\bar{\mathbb{P}}_n \Delta_{R,\widehat{\eta}} \leq 0$.

Next we bound $\bar{\mathbb{P}} \Delta_{R,\widehat{\eta}} - \bar{\mathbb{P}}_n \Delta_{R,\widehat{\eta}}$. Since $|y - f(\boldsymbol{x})| \in [0, 1], \eta(\boldsymbol{x}) \in [0, 1]$, it can be easily verified that $\mathrm{Var}_{\boldsymbol{x} \sim \mathbb{P}}[\Delta_{R,\eta}] \leq 2\mathbb{E}_{\boldsymbol{x} \sim \mathbb{P}}[\Delta_{R,\eta}]$. To invoke Theorem 3.3 in Bartlett et al. (2005), we need to find a subroot function $\psi(r)$ such that

$$\psi(r) \geq 2\mathbb{E}\bar{\mathbb{P}}_n\{\Delta_{R,\widehat{\eta}} \in \mathcal{H} : \mathbb{E}[h^2] \leq r\}.$$

To this end, we show some analysis on the Local Rademacher Average $\mathbb{E}\mathfrak{R}_n\{\Delta_{R,\widehat{\eta}} \in \mathcal{H} : \mathbb{E}[h^2] \leq r\}$.

$$\mathbb{E}\mathfrak{R}_n(\Delta \circ R \circ \mathcal{G}, r) = \mathbb{E}_{S_n \sigma_{1:n}}\left[ \sup_{\eta \in \mathcal{G}, \mathbb{E}_{\boldsymbol{x}, y}\left[\Delta^2_{R,\widehat{\eta}}\right] \leq r} \frac{1}{n} \sum_{i=1}^n \sigma_i \Delta_{R,\widehat{\eta}} \right].$$

The following analysis largely follows from the proof of Corollary 3.7 in Bartlett et al. (2005). Since $\Delta_{R,\widehat{g}}$ is uniformly bounded by 2, for any $r \geq \psi(r)$, Corollary 2.2 in Bartlett et al. (2005) implies that with probability at least $1 - \frac{1}{n}$, $\{h \in *\mathcal{H} : \bar{\mathbb{P}}h^2 \leq r\} \subseteq \{h \in *\mathcal{H} : \bar{\mathbb{P}}_n h^2 \leq 2r\}$. Let $\mathcal{E} \triangleq \{h \in *\mathcal{H} : \mathbb{P}h^2 \leq r\} \subseteq \{h \in *\mathcal{H} : \bar{\mathbb{P}}_n h^2 \leq 2r\}$, then the following holds:

$$\mathbb{E}\mathfrak{R}_n\{*\mathcal{H}, \bar{\mathbb{P}}h^2 \leq r\} \leq \mathbb{P}[\mathcal{E}]\mathbb{E}[\mathfrak{R}_n\{*\mathcal{H}, \bar{\mathbb{P}}h^2 \leq r\}|\mathcal{E}] + \mathbb{P}[\mathcal{E}^c]\mathbb{E}[\mathfrak{R}_n\{*\mathcal{H}, \bar{\mathbb{P}}h^2 \leq r\}|\mathcal{E}^c]$$

$$\leq \mathbb{E}[\mathfrak{R}_n\{*\mathcal{H}, \bar{\mathbb{P}}_n h^2 \leq 2r\}] + \frac{2}{n}.$$

Since $r^* = \psi(r^*)$, $r^*$ satisfies the following

$$r^* \leq 20B\mathbb{E}\mathfrak{R}_n\{*\mathcal{H}, \bar{\mathbb{P}}_n h^2 \leq 2r^*\} + \frac{22 \log n}{n}. \tag{21}$$

Next we leverage Dudley's entropy integral (Dudley, 2014) to upper bound $\mathbb{E}\mathfrak{R}_n\{*\mathcal{H}, \bar{\mathbb{P}}_n h^2 \leq 2r^*\}$, using the integral of covering number. Specifically, by applying the chaining bound, it follows from Theorem B.7 (Bartlett et al., 2005) that

$$\mathbb{E}[\mathfrak{R}_n(*\mathcal{H}, \bar{\mathbb{P}}_n h^2 \leq 2r^*)] \leq \frac{\mathrm{const}}{\sqrt{n}} \mathbb{E} \int_0^{\sqrt{2r^*}} \sqrt{\log \mathcal{N}_2(\varepsilon, *\mathcal{H}, \boldsymbol{x}_{1:n})} d\varepsilon, \tag{22}$$

where const represents some universal constant. Next we bound the covering number $\log \mathcal{N}_2(\varepsilon, \mathcal{H}, \boldsymbol{x}_{1:n})$ by $\log \mathcal{N}_2(\varepsilon, \mathcal{G}, \boldsymbol{x}_{1:n})$. We show that for all $\boldsymbol{x}_{1:n}$, any $\varepsilon^2$-cover of $\mathcal{G}$ is a $\varepsilon$-cover of $\mathcal{H}$. Specifically, let $\mathcal{V} \subset [0, 1]^n$ be an $\varepsilon$-cover of $\mathcal{H}$ on $\boldsymbol{x}_{1:n}$ so that for all $\eta \in \mathcal{G}$, $\exists \boldsymbol{v}_{1:n} \in \mathcal{V}$ so that $\sqrt{\frac{1}{n} \sum_{i \in [n]} (\eta(\boldsymbol{x}_i) - v_i)^2} \leq \varepsilon$. Now we show

that for any $\boldsymbol{z}_{1:n}$, the family of $(\boldsymbol{v}_i - y_i)^2 - (\eta_i^* - y_i)^2, i \in [n]$ is an $\varepsilon$-cover for $\mathcal{H}$, where $\eta_i^* \triangleq \mathbb{P}(y_i = 1|\boldsymbol{x}_i)$:

$$
\sqrt{\frac{1}{n}\sum_{i\in[n]}((\boldsymbol{v}_i - y_i)^2 - (\eta^* - y_i)^2 - \Delta_{R,\eta})^2} = \sqrt{\frac{1}{n}\sum_{i\in[n]}((\boldsymbol{v}_i - y_i)^2 - (\eta(\boldsymbol{x}_i) - y_i)^2)^2}
$$

$$
= \sqrt{\frac{1}{n}\sum_{i\in[n]}(\boldsymbol{v}_i - \eta(\boldsymbol{x}_i))^2(\boldsymbol{v}_i + \eta(\boldsymbol{x}_i) - 2y_i)^2} \le 4\varepsilon.
$$

Combine the above inequality with Corollary 3.7 from Bartlett et al. (2005), we have the following upper bound on the entropy number of $*\mathcal{H}$:

$$
\log \mathcal{N}_2(\varepsilon, *\mathcal{H}, \boldsymbol{x}_{1:n}) \le \log\left\{\mathcal{N}_2\left(\frac{\varepsilon}{2}, \mathcal{H}, \boldsymbol{x}_{1:n}\right)\left(\lceil\frac{2}{\varepsilon}\rceil + 1\right)\right\} \le \log\left\{\mathcal{N}_2\left(\frac{\varepsilon}{2}, \mathcal{G}, \boldsymbol{x}_{1:n}\right)\left(\lceil\frac{2}{\varepsilon}\rceil + 1\right)\right\}.
$$

Next we bound $\frac{\text{const}}{\sqrt{n}}\mathbb{E}\int_0^{\sqrt{2r^*}}\sqrt{\log \mathcal{N}_2(\varepsilon, *\mathcal{H}, \boldsymbol{x}_{1:n})}d\varepsilon$ from (22). Note that by Haussler's covering number bound (Haussler, 1995), the following inequality holds: $\log \mathcal{N}_2\left(\frac{\varepsilon}{8}, \mathcal{G}, \boldsymbol{x}_{1:n}\right) \le cd_P\log\left(\frac{1}{\varepsilon}\right)$, we can therefore derive the following inequalities:

$$
\frac{\text{const}}{\sqrt{n}}\mathbb{E}\int_0^{\sqrt{2r^*}}\sqrt{\log \mathcal{N}_2(\varepsilon, *\mathcal{H}, \boldsymbol{x}_{1:n})}d\varepsilon \le \frac{\text{const}}{\sqrt{n}}\mathbb{E}\int_0^{\sqrt{2r^*}}\sqrt{\log \mathcal{N}_2\left(\frac{\varepsilon}{2}, \mathcal{H}, \boldsymbol{x}_{1:n}\right)\left(\lceil\frac{2}{\varepsilon}\rceil + 1\right)}d\varepsilon
$$

$$
\le \frac{\text{const}}{\sqrt{n}}\mathbb{E}\int_0^{\sqrt{2r^*}}\sqrt{\log \mathcal{N}_2\left(\frac{\varepsilon}{8}, \mathcal{G}, \boldsymbol{x}_{1:n}\right)\left(\lceil\frac{2}{\varepsilon}\rceil + 1\right)}d\varepsilon
$$

$$
\le \text{const}\sqrt{\frac{d_P(\mathcal{G})}{n}}\int_0^{\sqrt{2r^*}}\sqrt{\log\left(\frac{1}{\varepsilon}\right)} \le \text{const}\sqrt{\frac{d_P(\mathcal{G})r^*\log(1/r^*)}{n}}
$$

$$
\le \text{const}\sqrt{\frac{d_P^2(\mathcal{G})}{n^2} + \frac{d_P(\mathcal{G})r^*\log(n/ed_P(\mathcal{G}))}{n}},
$$

where const represents some universal constants that may change from line to line. Together with (21) one can solve for

$$
r^* \lesssim \frac{d_P(\mathcal{G})\log(\frac{n}{d_P(\mathcal{G})})}{n}.
$$

Since $\bar{\mathbb{P}}\Delta_{R,\eta} \le \varepsilon$, we have $\mathbb{P}|\widehat{\eta} - \eta^*|^2 \le \varepsilon$, given that the following equality holds:

$$
\bar{\mathbb{P}}\Delta_{R,\eta} = \mathbb{E}_{\boldsymbol{z}}[R(\widehat{\eta}; f^*, \boldsymbol{z}) - R(\eta^*; f^*, \boldsymbol{z})]
$$

$$
= \mathbb{E}_{\boldsymbol{x},y}[(\widehat{\eta}(\boldsymbol{x}) - y)^2 - (\eta^*(\boldsymbol{x}) - y)^2]
$$

$$
= \mathbb{E}_{\boldsymbol{x}}[|\widehat{\eta}(\boldsymbol{x}) - \eta^*(\boldsymbol{x})|^2]. \qquad (\text{as } \eta^*(\boldsymbol{x}) \triangleq \mathbb{E}[y])
$$

Note that since $\left||\widehat{\eta} - \frac{1}{2}| - |\eta^* - \frac{1}{2}|\right|^2 \le |\widehat{\eta} - \eta^*|^2$, the following readily follows:

$$
\mathbb{E}_{\boldsymbol{x}}\left[(|\widehat{\eta}(\boldsymbol{x}) - \frac{1}{2}| - |\eta^*(\boldsymbol{x}) - \frac{1}{2}|)^2\right] \le \varepsilon. \tag{23}
$$

$\square$

## A.6 Proof of Theorem 4.4

*Proof.* The proof is by construction. We construct the full support of $\boldsymbol{x}$ to be $\mathcal{X} = \mathcal{X}_1 \cup \mathcal{X}_2$. For all $\boldsymbol{x} \in \mathcal{X}_1$, let $\omega^*(\boldsymbol{x}) = |\eta^*(\boldsymbol{x}) - \frac{1}{2}| \ge c$ and $\boldsymbol{x} \in \mathcal{X}_2$, let $\omega^*(\boldsymbol{x}) = |\eta^*(\boldsymbol{x}) - \frac{1}{2}| = 0$. By decomposing the excessive risk we

have:

$$\mathbb{E}_{\boldsymbol{x},y}[\mathbb{1}\{\widehat{f} \neq y\} - \mathbb{1}\{f^* \neq y\}] = \mathbb{E}_{\boldsymbol{x},y}[\mathbb{1}\{\widehat{f} \neq y\} - \mathbb{1}\{f^* \neq y\}|\boldsymbol{x} \in \mathcal{X}_1] \cdot \mathbb{P}[\boldsymbol{x} \in \mathcal{X}_2]$$
$$+ \underbrace{\mathbb{E}_{\boldsymbol{x},y}[\mathbb{1}\{\widehat{f} \neq y\} - \mathbb{1}\{f^* \neq y\}|\boldsymbol{x} \in \mathcal{X}_2] \cdot \mathbb{P}[\boldsymbol{x} \in \mathcal{X}_2]}_{=0,\text{since for all } \boldsymbol{x} \in \mathcal{X}_2, \, \mathbb{P}(y=1|\boldsymbol{x}) = \mathbb{P}(y=-1|\boldsymbol{x})}.$$

The lower bound of term $\mathbb{E}_{\boldsymbol{x},y}[\mathbb{1}\{\widehat{f} \neq y\} - \mathbb{1}\{f^* \neq y\}|\boldsymbol{x} \in \mathcal{X}_1] \geq \frac{1}{cn}$ could be established by Theorem 4 in Massart & Nédélec (2006). $\qquad\square$

## A.7 Proof of Theorem 4.2

Throughout this proof, we use superscript $j$ to index the $j$th coordinate of a vector $\boldsymbol{x}$ (e.g., $\boldsymbol{x}^j$), and this is to be distinguished from the subscript $i$ that indexes the samples.

*Proof.* The proof is by construction. Let $\gamma$ be the minimum margin. Let $\mathcal{F} = \{\mathbf{1}^\top \text{sign}(\boldsymbol{x} - \beta)|\beta \in \mathbb{R}^d\}, \mathcal{X} \equiv \bigcup_{j=1}^d \mathcal{E}^j, \mathcal{E}^j \equiv \alpha \cdot \boldsymbol{e}^j, \alpha \in \{[-2,-1] \cup \{-0.1\} \cup [1,2] \cup \{0.1\}\}, j \in [d]$ where $\boldsymbol{e}^j$ is $j$-th standard basis. Let

$$\eta^*(x) = \frac{1}{2}\mathbb{1}\{\mathbf{1}^\top \text{sign}(\boldsymbol{x}) \geq 0\}\left\{1 + \left\{\mathbb{1}\{\|\boldsymbol{x}\| = 0.1\} + \mathbb{1}\{\|\boldsymbol{x}\| \geq 1\}\gamma\right\}\right\} \tag{24}$$

Note $\mathbf{1}^\top \text{sign}(\boldsymbol{x} - \beta)$ could be viewed as a composition class of the $d$-dimensional half-space and the interval function, and its VC-dimension could be bounded as $d\log(d)$ (Vidyasagar, 2013). Consequently, the choice of $n$ implies that $\mathbb{E}_{\boldsymbol{x}}[\mathbb{1}\{\widehat{f}_{\text{wERM}} \neq f^*\}|\omega^*(\boldsymbol{x}) > \gamma] \lesssim \varepsilon$ for any given $\widehat{\omega}$ that satisfies $\mathbb{E}_{\boldsymbol{x}}[(\omega(\boldsymbol{x}) - \omega^*(\boldsymbol{x}))^2] \leq \varepsilon$.

Consider following data generative process:

$$j \sim \text{Unif}\{1, 2, ..., d\}$$

$$\alpha = \begin{cases} 0.1, & \text{with prob } \frac{\gamma}{32} \\ -0.1, & \text{with prob } \frac{\gamma}{32} \\ \text{Unif}(1, 2), & \text{with prob } 1 - \frac{3\gamma}{32} \\ \text{Unif}(-2, -1), & \text{with prob } \frac{\gamma}{32} \end{cases}$$

$$\boldsymbol{x} = \alpha \cdot \boldsymbol{e}^j$$

$$y = \begin{cases} 1, & \text{with prob } \eta^*(\boldsymbol{x}) \\ -1 & \text{with prob } 1 - \eta^*(\boldsymbol{x}) \end{cases}$$

Note that the following version of the Chernoff inequality will be applied multiple times in the proof:

$$\mathbb{P}[|\sum_i^m X_i - m\mathbb{E}[X]| \geq \xi m\mathbb{E}[X]] \leq 2e^{-\xi^2 m\mathbb{E}X/3}. \tag{25}$$

By setting $\xi = 0.5$ in inequality (25) and taking a union bound over $\mathcal{E}^{1:d}$ we have

$$\mathbb{P}\left[\exists j, s.t., \left||\mathcal{E}^j \bigcap S_n| - \frac{64\log d\log(1/\delta)}{\gamma\varepsilon}\right| \geq \frac{32\log d\log(1/\delta)}{\gamma\varepsilon}\right]$$
$$\leq 2de^{-5\log d\log(1/\delta)} \leq 2de^{-5\log(d/\delta)} \leq \delta. \tag{26}$$

The inequality above implies that with probability at least $1 - \delta$, $\forall j \in [d]$, we have

$$\frac{32d\log(d)\log(1/\delta)}{\gamma\varepsilon} \leq |\mathcal{E}^j \bigcap S_n| \leq \frac{96d\log(d)\log(1/\delta)}{\gamma\varepsilon}. \tag{27}$$

Next we show that for each $j \in [d]$, for sufficiently small $\gamma$ with constant probability, $\widehat{\beta}_{\text{ERM}}^j \neq \beta^*$. We first present our argument for the case where $d = 1$. W.O.L.G, we focus on the case where $\mathcal{X} = \mathcal{X}^1$. Given

$\boldsymbol{x}_{1:n}^1$, let $\boldsymbol{x}_{(-n_1):(-1)}^1 \cup \boldsymbol{x}_{(1):(n_2):(n_2+n_3)}^1$ be an ordering of $\boldsymbol{x}_n$ with $\boldsymbol{x}_{-i}^1 \leq \boldsymbol{x}_{-i+1}^1 \leq \boldsymbol{x}_{-1}^1 < 0 < \boldsymbol{x}_1^1 \leq \boldsymbol{x}_{n_2}^1 < \boldsymbol{x}_{n_2+i}^1 < \boldsymbol{x}_{n_2+n_3}^1 < \boldsymbol{x}_{n_3}^1$, where $\boldsymbol{x}_{(1):(n_2)} = 0.1$ and $n_1 + n_2 + n_3 = n$. Since $\frac{32\log(1/\delta)}{\gamma\varepsilon} \leq n \leq \frac{96\log(1/\delta)}{\gamma\varepsilon}$, by Chernoff inequality in (25) with $\xi = 0.5$ and picking $\tau = \gamma$, we have $\frac{\log(1/\delta)}{\varepsilon} \leq n_2 \leq \frac{3\log(1/\delta)}{\varepsilon}$ and $\frac{\tau 32\log(1/\delta)}{\gamma\varepsilon} \leq \tau n_3 \leq \frac{\tau 96\log(1/\delta)}{\gamma\varepsilon}$ that hold simultaneously with probability at least $1 - 2\delta$. Consider $\tau$ fraction of positive samples: $\boldsymbol{x}_{(n_2+1):(\lfloor \tau n_3 \rfloor)}$. Given $n_2$, by Lemma 2, we have ,

$$\mathbb{P}\left[\sum_{i=n_2+1}^{(n_2+\lfloor \tau n_3 \rfloor)} \mathbb{1}\{y_{(i)} = f^*(\boldsymbol{x}_{(i)})\} \leq (\frac{1}{2}+\gamma)(1-\xi)\tau n_3\right] \geq 0.2e^{-6\xi^2 \tau n_3 (\frac{1}{2}+\gamma)}. \tag{28}$$

By inequality (28) with $\xi = 3\gamma$, we have that with probability at least $0.2e^{-\frac{54\gamma^2 \log(d)\log(1/\delta)(\frac{1}{2}+\gamma)}{\varepsilon}}$,

$$\sum_{i=n_2+1}^{(n_2+\lfloor \tau n_3 \rfloor)} \mathbb{1}\{y_{(i)} = f^*(\boldsymbol{x}_{(i)})\} \leq (\frac{1}{2}+\gamma)(1-3\gamma)n_3$$

$$\implies \sum_{i=n_2+1}^{(n_2+\lfloor \tau n_3 \rfloor)} \mathbb{1}\{y_{(i)} = f^*(\boldsymbol{x}_{(i)})\} \leq (1+2\gamma)(1-3\gamma)\frac{n_3}{2}$$

$$(\text{set } \gamma < \frac{1}{12}) \implies \sum_{i=n_2+1}^{(n_2+\lfloor \tau n_3 \rfloor)} \mathbb{1}\{y_{(i)} = f^*(\boldsymbol{x}_{(i)})\} \leq (1-\frac{\gamma}{2})\frac{n_3}{2}.$$

Therefore,

$$\sum_{i=n_2+1}^{n_2+(\lfloor \tau n_3 \rfloor)} \mathbb{1}\{y_{(i)} = f^*(\boldsymbol{x}_{(i)})\} + \sum_{i=1}^{n_2} \mathbb{1}\{y_{(i)} = f^*(\boldsymbol{x}_{(i)})\}$$

$$\leq (1-\frac{\gamma}{2})\frac{n_3}{2} + \frac{\gamma n_3}{4} \leq \frac{n_3}{2} \tag{29}$$

$$\leq \sum_{i=n_2+1}^{(n_2+\lfloor \tau n_3 \rfloor)} \mathbb{1}\{y_{(i)} \neq f^*(\boldsymbol{x}_{(i)})\}.$$

It can be easily verified that inequality (29) implies that $\widehat{\beta^j}_{\text{ERM}} \geq \boldsymbol{x}_{(\tau n_3)}$. To ensure that

$$0.2e^{-\frac{54\gamma^2 \log(d)\log(1/\delta)(\frac{1}{2}+\gamma)}{\varepsilon}} \geq 0.12,$$

it suffices to pick $\gamma = \sqrt{\frac{\log(d)\log(1/\delta)}{55\varepsilon}}$. Since with probability 0.12 we have $\widehat{\beta}_{\text{ERM}}^j \geq \boldsymbol{x}_{(\tau n_3)} > 0.1$, which implies that $\mathbb{E}_{\boldsymbol{x}}[\mathbb{1}\{\widehat{f}_{\text{ERM}} \neq f^*\}|\omega^*(\boldsymbol{x}) > \gamma, \boldsymbol{x} \in \mathcal{E}^1] \geq 0.5$. With markov inequality, we have with probability at least 0.1, 0.03 fraction of $\mathcal{E}^{1:d}$ has $\widehat{\beta}_{ERM}^j > 0.1$, which implies that $\mathbb{E}_{\boldsymbol{x}}[\mathbb{1}\{\widehat{f}_{\text{ERM}} \neq f^*\}|\omega^*(\boldsymbol{x}) > \gamma, \boldsymbol{x} \in \mathcal{E}^j] \geq 0.5$. We have with probability at least 0.1, $\mathbb{E}_{\boldsymbol{x}}[\mathbb{1}\{\widehat{f}_{\text{ERM}} \neq f^*\}|\omega^*(\boldsymbol{x}) > \gamma] \geq 0.015$. $\qquad \square$

## A.8 Proof of Theorem 4.5

The proof readily follows from Theorem 4.7 with $\ell(f; \boldsymbol{z}) \triangleq (y - f(\boldsymbol{x}))^2$, $\omega^*(\boldsymbol{x}) = \frac{C}{\sigma^{2*}(\boldsymbol{x})}$ and $\mathcal{D}(f^*(\boldsymbol{x}), \widehat{f}(\boldsymbol{x})) = \frac{\sigma^{2*}(\boldsymbol{x})}{C}(f^* - \widehat{f})^2$. It can be easily verified that Assumption 1 is satisfied with $a = 8 + \frac{8c_2^2}{\sqrt{\gamma}}, b = \frac{4}{C\gamma}, L = \frac{c_2}{\sqrt{\gamma}}$.

## A.9 Proof of Theorem 4.6 and discussion

Our analysis of learning $\widehat{\sigma^2}(\boldsymbol{x})$ is based on the following negative log-likelihood loss:

$$\ell_{\text{NLL}}(\sigma^2, f, \boldsymbol{z}) \triangleq \log(\sigma^2(\boldsymbol{x})) + \frac{(y-f(\boldsymbol{x}))^2}{\sigma^2(\boldsymbol{x})}. \tag{30}$$

In particular, we seek to minimize risk of the form in Equation (30) while restricting $f$ and $\sigma^2$ to be in hypothesis classes $\widetilde{\mathcal{F}} \subseteq \mathcal{F}, \widetilde{\mathcal{G}} \subseteq \mathcal{G}$ that satisfy $\frac{(y-f(\boldsymbol{x}))^2}{\sigma^2(\boldsymbol{x})} \leq 4c_2^2$, uniformly for all $\boldsymbol{z} = (\boldsymbol{x}, y)$. For learning $\sigma^{2*}(\boldsymbol{x})$. For simplicity, we assume $\mathbb{E}_\xi[(\xi^2 - 1)^2]$ being a constant.

**Verification of Bernstein-type condition** We show that the risk function in Equation (30) satisfies some Bernstein-type condition. Since $\frac{(y-f(\boldsymbol{x}))^2}{\sigma^2(\boldsymbol{x})} \leq 4c_2^2$, it then follows that $|\ell_{\mathrm{NLL}}(\sigma^2, f, \boldsymbol{z}) - \ell_{\mathrm{NLL}}(\sigma^{2*}, f^*, \boldsymbol{z})| \leq 5c_2^2 + 2\log(\frac{c_2}{\gamma})$. In following analysis, we let $C = 5c_2^2 + 2\log(\frac{c_2}{\gamma})$. The expectation term then satisfies:

$$
\mathbb{E}_{\boldsymbol{z}}[\ell_{\mathrm{NLL}}(\sigma^2, f, \boldsymbol{z}) - \ell_{\mathrm{NLL}}(\sigma^{2*}, f^*, \boldsymbol{z})]
$$

$$
= \mathbb{E}_{\boldsymbol{x}}\left[ \frac{(f(\boldsymbol{x}) - f^*(\boldsymbol{x}))^2}{\sigma^2(\boldsymbol{x})} - 1 + \frac{\sigma^{2*}(\boldsymbol{x})}{\sigma^2(\boldsymbol{x})} - \log\left(\frac{\sigma^{2*}(\boldsymbol{x})}{\sigma^2(\boldsymbol{x})}\right) \right] \tag{31}
$$

$$
\geq \mathbb{E}_{\boldsymbol{x},\xi}\left[ \frac{(f(\boldsymbol{x}) - f^*(\boldsymbol{x}))^2}{\sigma^2(\boldsymbol{x})} + \frac{\left(\frac{\sigma^{2*}(\boldsymbol{x})}{\sigma^2(\boldsymbol{x})} - 1\right)^2}{2} \cdot \frac{\xi^4}{\max\{\xi^4, \left(\xi^2 \frac{\sigma^{2*}(\boldsymbol{x})}{\sigma^2(\boldsymbol{x})}\right)^2\}} \right]; \tag{32}
$$

the last inequality leverages the fact that for all $t > 0$

$$
\log\left(\frac{1}{t}\right) \geq 1 - t + \frac{(t-1)^2}{2\max(1, t)}. \tag{33}
$$

For the variance term, the following holds:

$$
\mathbf{Var}_{\boldsymbol{z}}[(\ell_{\mathrm{NLL}}(\sigma^2, f, \boldsymbol{z}) - \ell_{\mathrm{NLL}}(\sigma^{2*}, f^*, \boldsymbol{z}))]
$$

$$
= \mathbb{E}_{\boldsymbol{z}}[(\ell_{\mathrm{NLL}}(\sigma^2, f, \boldsymbol{z}) - \ell_{\mathrm{NLL}}(\sigma^{2*}, f^*, \boldsymbol{z}))^2] - \mathbb{E}_{\boldsymbol{z}}[(\ell_{\mathrm{NLL}}(\sigma^2, f, \boldsymbol{z}) - \ell_{\mathrm{NLL}}(\sigma^{2*}, f^*, \boldsymbol{z}))]^2
$$

$$
\leq \mathbb{E}_{\boldsymbol{x},\xi}\left[ \left( \frac{(f(\boldsymbol{x}) - f^*(\boldsymbol{x}))^2}{\sigma^2(\boldsymbol{x})} - \xi^2 + \frac{\xi^2 \sigma^{2*}(\boldsymbol{x})}{\sigma^2(\boldsymbol{x})} - \frac{2\xi\sqrt{\sigma^{2*}(\boldsymbol{x})}(f(\boldsymbol{x}) - f^*(\boldsymbol{x}))}{\sigma^2(\boldsymbol{x})} - \log\left(\frac{\sigma^{2*}(\boldsymbol{x})}{\sigma^2(\boldsymbol{x})}\right) \right)^2 \right];
$$

this can be further decomposed as

$$
\mathbb{E}_{\boldsymbol{x},\xi}\left[ \left( (\xi^2 - 1)\left(\frac{\sigma^{2*}(\boldsymbol{x})}{\sigma^2(\boldsymbol{x})} - 1\right) - \frac{2\xi\sqrt{\sigma^{2*}(\boldsymbol{x})}(f(\boldsymbol{x}) - f^*(\boldsymbol{x}))}{\sigma^2(\boldsymbol{x})} \right)^2 \right]
$$

$$
+ \mathbb{E}_{\boldsymbol{x}}\left[ \left( \frac{(f(\boldsymbol{x}) - f^*(\boldsymbol{x}))^2}{\sigma^2(\boldsymbol{x})} - 1 + \frac{\sigma^{2*}(\boldsymbol{x})}{\sigma^2(\boldsymbol{x})} - \log\left(\frac{\sigma^{2*}(\boldsymbol{x})}{\sigma^2(\boldsymbol{x})}\right) \right)^2 \right]
$$

$$
+ 2\mathbb{E}_{\boldsymbol{x},\xi}\left[ \left( \frac{(f(\boldsymbol{x}) - f^*(\boldsymbol{x}))^2}{\sigma^2(\boldsymbol{x})} - 1 + \frac{\sigma^{2*}(\boldsymbol{x})}{\sigma^2(\boldsymbol{x})} - \log\left(\frac{\sigma^{2*}(\boldsymbol{x})}{\sigma^2(\boldsymbol{x})}\right) \right) \bullet \right.
$$

$$
\left. \left( (\xi^2 - 1)\left(\frac{\sigma^{2*}(\boldsymbol{x})}{\sigma^2(\boldsymbol{x})} - 1\right) - \frac{2\xi\sqrt{\sigma^{2*}(\boldsymbol{x})}(f(\boldsymbol{x}) - f^*(\boldsymbol{x}))}{\sigma^2(\boldsymbol{x})} \right) \right]
$$

$$
\leq \underbrace{2\mathbb{E}_{\boldsymbol{x},\xi}\left[ \left( (\xi^2 - 1)^2 \left(\frac{\sigma^{2*}(\boldsymbol{x})}{\sigma^2(\boldsymbol{x})} - 1\right)^2 \right) \right]}_{\text{Term I}} + \underbrace{2\mathbb{E}_{\boldsymbol{x},\xi}\left[ \left( \frac{\xi\sqrt{\sigma^{2*}(\boldsymbol{x})}(f(\boldsymbol{x}) - f^*(\boldsymbol{x}))}{\sigma^2(\boldsymbol{x})} \right)^2 \right]}_{\text{Term II}}
$$

$$
+ \underbrace{C\mathbb{E}_{\boldsymbol{z}}[\ell_{\mathrm{NLL}}(\sigma^2, f, \boldsymbol{z}) - \ell_{\mathrm{NLL}}(\sigma^{2*}, f^*, \boldsymbol{z})]}_{\text{Term III}}. \tag{34}
$$

**Bounding Term I:** Due to the fact that for all $t > 0$, $\max\{2, 2t\}(-\log(t) - 1 + t) \geq (t-1)^2$, the following holds:

$$
\mathbb{E}_{\boldsymbol{x}}\left[ 2\max\left\{1, \frac{\sigma^{2*}(\boldsymbol{x})}{\sigma^2(\boldsymbol{x})}\right\}\left( -1 + \frac{\sigma^{2*}(\boldsymbol{x})}{\sigma^2(\boldsymbol{x})} - \log\left(\frac{\sigma^{2*}(\boldsymbol{x})}{\sigma^2(\boldsymbol{x})}\right) \right) \right]
$$

$$
= \mathbb{E}_{\boldsymbol{x},\xi}\left[ 2\xi^2 \max\left\{1, \frac{\sigma^{2*}(\boldsymbol{x})}{\sigma^2(\boldsymbol{x})}\right\}\left( -1 + \frac{\sigma^{2*}(\boldsymbol{x})}{\sigma^2(\boldsymbol{x})} - \log\left(\frac{\sigma^{2*}(\boldsymbol{x})}{\sigma^2(\boldsymbol{x})}\right) \right) \right] \geq \mathbb{E}_{\boldsymbol{x}}\left[ \left(\frac{\sigma^{2*}(\boldsymbol{x})}{\sigma^2(\boldsymbol{x})} - 1\right)^2 \right].
$$

On the other hand, since for all $f \in \widetilde{\mathcal{F}}, \sigma^2 \in \widetilde{\mathcal{G}}, \frac{(y-f(\boldsymbol{x}))^2}{\sigma^2(\boldsymbol{x})} \leq 4c_2^2$, we further have $\frac{\xi^2 \sigma^{2*}(\boldsymbol{x})}{\sigma^2(\boldsymbol{x})} \leq 2c_2^2 + 4/c_3^2$. Term I could be bounded as:

$$2\mathbb{E}_{\boldsymbol{x},\xi}\left[\left((\xi^2-1)^2\left(\frac{\sigma^{2*}(\boldsymbol{x})}{\sigma^2(\boldsymbol{x})}-1\right)^2\right] \leq \left(4c_2^2 + \frac{4}{c_3^2}\right)\mathbb{E}_{\boldsymbol{x},\xi}\left[-1+\frac{\sigma^{2*}(\boldsymbol{x})}{\sigma^2(\boldsymbol{x})} - \log\left(\frac{\sigma^{2*}(\boldsymbol{x})}{\sigma^2(\boldsymbol{x})}\right)\right].$$

**Bounding Term II:** Since for all $f \in \widetilde{\mathcal{F}}, \sigma^2 \in \widetilde{\mathcal{G}}, \frac{(y-f(\boldsymbol{x}))^2}{\sigma^2(\boldsymbol{x})} \leq 4c_2^2$, we further have $\sqrt{\frac{\xi \sigma^{2*}(\boldsymbol{x})}{\sigma^2(\boldsymbol{x})}} \leq 2c_2 + 2/c_3$. Consequently, we have

$$2\mathbb{E}_{\boldsymbol{x},\xi}\left[\left(\frac{\xi\sqrt{\sigma^{2*}(\boldsymbol{x})}(f(\boldsymbol{x})-f^*(\boldsymbol{x}))}{\sigma^2(\boldsymbol{x})}\right)^2\right] \leq \left(4c_2^2 + \frac{4}{c_3^2}\right)\mathbb{E}_{\boldsymbol{x}}\left[\frac{(f(\boldsymbol{x})-f^*(\boldsymbol{x}))^2}{\sigma^2(\boldsymbol{x})}\right].$$

As a result, the following holds:

$$\mathbf{Var}_{\boldsymbol{z}}[(\ell_{\mathrm{NLL}}(\sigma^2, f, \boldsymbol{z}) - \ell_{\mathrm{NLL}}(\sigma^{2*}, f^*, \boldsymbol{z}))]$$
$$\lesssim \left(C + c_2^2 + \frac{1}{c_3^2}\right)\mathbb{E}_{\boldsymbol{z}}[\ell_{\mathrm{NLL}}(\sigma^2, f, \boldsymbol{z}) - \ell_{\mathrm{NLL}}(\sigma^{2*}, f^*, \boldsymbol{z})].$$

Next, we move on to the proof of the theorem statement.

*Proof.* Let $B = \left(C + c_2^2 + \frac{1}{c_3^2}\right)$, the Bernstein constant. Define the following function class:

$$\mathcal{H} \equiv \Delta \circ L \circ \widetilde{\mathcal{F}} \times \widetilde{\mathcal{G}} \equiv \left\{\Delta\ell_{NLL}(\sigma^2, f; \sigma^{2*}, f^*, \boldsymbol{z}) = \ell_{\mathrm{NLL}}(\sigma^2, f, \boldsymbol{z}) - \ell_{\mathrm{NLL}}(\sigma^{2*}, f^*, \boldsymbol{z}) : f \in \widetilde{\mathcal{F}}, \sigma^2 \in \widetilde{\mathcal{G}}\right\}$$

which is a composite function of hypothesis class $\widetilde{\mathcal{F}} \times \widetilde{\mathcal{G}}$, loss function $L$ and the difference operation $\Delta$. For simplicity we let $\Delta_{\ell,\sigma^2,f} = \Delta\ell_{\mathrm{NLL}}(\widehat{\sigma}^2, \widehat{f}; \sigma^{2*}, f^*, \boldsymbol{z})$ when there is no ambiguity. By definition of $(\widehat{f}, \widehat{\sigma^2})$, we have $\bar{\mathbb{P}}_n\ell_{\mathrm{NLL}}(\widehat{\sigma}^2, \widehat{f}, \boldsymbol{z}) \leq \bar{\mathbb{P}}_n\ell_{\mathrm{NLL}}(\sigma^{2*}, f^*, \boldsymbol{z})$ and therefore $\bar{\mathbb{P}}_n\Delta_{\ell,\sigma^2,f} \leq 0$.

Next we bound $\bar{\mathbb{P}}\Delta_{\ell,\sigma^2,f} - \bar{\mathbb{P}}_n\Delta_{\ell,\sigma^2,f}$. According to (34), we have $\mathrm{Var}_{\boldsymbol{z}}[\Delta_{\ell,\sigma^2,f}] \leq B\mathbb{E}_{\boldsymbol{z}}[\Delta_{\ell,\sigma^2,f}]$. To invoke Theorem 3.3 in Bartlett et al. (2005), we need to find a sub-root function $\psi(r)$ such that

$$\psi(r) \geq 2B\mathbb{E}\bar{\mathbb{P}}_n\{\Delta_{\ell,\sigma^2,f} \in \mathcal{H} : \mathbb{E}[h^2] \leq r\}.$$

To find $\psi(r)$, we show some analysis on the Local Rademacher Average $\mathbb{E}\mathfrak{R}_n\{\Delta_{\ell,\sigma^2,f} \in \mathcal{H} : \mathbb{E}[h^2] \leq r\}$. Note that

$$\mathbb{E}\mathfrak{R}_n(\mathcal{H}, r) = \mathbb{E}_{S_n\sigma_{1:n}}\left[\sup_{f\in\widetilde{\mathcal{F}},\sigma^2\in\widetilde{\mathcal{G}},\mathbb{E}_{\boldsymbol{x},y}[\Delta_{\ell,\sigma^2,f}]\leq r} \frac{1}{n}\sum_{i=1}^n \sigma_i\Delta_{\ell,\sigma^2,f}\right],$$

and we have $h(\boldsymbol{x}) \in [-C, C]$. By leveraging some analysis from the proof in Corollary 3.7 in Bartlett et al. (2005), we bound $\{h \in *\mathcal{H} : \bar{\mathbb{P}}h^2 \leq r\}$ using $\{h \in *\mathcal{H} : \bar{\mathbb{P}}_n h^2 \leq r\}$ where the latter could be applied in the entropy integral. Since $\Delta_{\ell,\sigma^2,f}$ is uniformly bounded by $2C$, for any $r \geq \psi(r)$, Corollary 2.2 in Bartlett et al. (2005) implies that with probability at least $1 - \frac{1}{n}$, $\{h \in *\mathcal{H} : \bar{\mathbb{P}}h^2 \leq r\} \subseteq \{h \in *\mathcal{H} : \bar{\mathbb{P}}_n h^2 \leq 2r\}$. Let $\mathcal{E}$ be event that $\{h \in *\mathcal{H} : \bar{\mathbb{P}}h^2 \leq r\} \subseteq \{h \in *\mathcal{H} : \bar{\mathbb{P}}_n h^2 \leq 2r\}$ holds, then we have

$$\mathbb{E}\mathfrak{R}_n\{*\mathcal{H}, \bar{\mathbb{P}}h^2 \leq r\} \leq \mathbb{P}[\mathcal{E}]\mathbb{E}[\mathfrak{R}_n\{*\mathcal{H}, \bar{\mathbb{P}}h^2 \leq r\}|\mathcal{E}] + \mathbb{P}[\mathcal{E}^c]\mathbb{E}[\mathfrak{R}_n\{*\mathcal{H}, \bar{\mathbb{P}}h^2 \leq r\}|\mathcal{E}^c]$$

$$\leq \mathbb{E}[\mathfrak{R}_n\{*\mathcal{H}, \bar{\mathbb{P}}_n h^2 \leq 2r\}] + \frac{2C^2}{n}.$$

Since $r^* = \psi(r^*)$, $r^*$ satisfies

$$r^* \leq 20BC\mathbb{E}\mathfrak{R}_n\{*\mathcal{H}, \bar{\mathbb{P}}_n h^2 \leq 2r^*\} + \frac{22C^2\log n}{n}. \tag{35}$$

Next we leverage the entropy integral (Dudley, 2014) to upper bound $\mathbb{E}\mathfrak{R}_n\{*\mathcal{H}, \bar{\mathbb{P}}_n h^2 \leq 2r^*\}$ using the integral of covering number. Apply the chaining bound, it follows from Theorem B.7 (Bartlett et al., 2005) that

$$\mathbb{E}[\mathfrak{R}_n(*\mathcal{H}, \bar{\mathbb{P}}_n h^2 \leq 2r^*)] \leq \frac{\text{const}}{\sqrt{n}}\mathbb{E}\int_0^{\sqrt{2r^*}} \sqrt{\log \mathcal{N}_2(\varepsilon, *\mathcal{H}, \boldsymbol{x}_{1:n})}d\varepsilon, \tag{36}$$

where const is some universal constant.

Next we bound the covering number $\log \mathcal{N}_2(\varepsilon, *\mathcal{H}, \boldsymbol{x}_{1:n})$ by $\log \mathcal{N}_2(\varepsilon, \widetilde{\mathcal{G}}, \boldsymbol{x}_{1:n}) + \log \mathcal{N}_2(\varepsilon, \widetilde{\mathcal{F}}, \boldsymbol{x}_{1:n})$. We show that for all $\boldsymbol{x}_{1:n}$, the composition of an $\varepsilon$-cover of $\widetilde{\mathcal{F}}$ and an $\varepsilon$-cover of $\widetilde{\mathcal{G}}$ gives rise to a an $\varepsilon$-cover of $\mathcal{H}$. Specifically, let $\mathcal{V} \subset [c_3, \frac{1}{\gamma}]^n$ be an $\varepsilon$-cover of $\widetilde{\mathcal{G}}$ on $\boldsymbol{x}_{1:n}$ so that for all $\sigma^2 \in \widetilde{\mathcal{G}}, \exists v_{1:n} \in \mathcal{V}$ so that $\sqrt{\frac{1}{n}\sum_{i\in[n]}(\sigma^2(\boldsymbol{x}_i) - v_i)^2} \leq \frac{\varepsilon}{4c_2^2/c_3 + 1/\gamma c_3}$, $\mathcal{U} \subset [-1, 1]^n$ be an $\varepsilon$-cover of $\widetilde{\mathcal{F}}$ on $\boldsymbol{x}_{1:n}$ so that for all $f \in \widetilde{\mathcal{F}}$, $\exists u_{1:n} \in \mathcal{U}$ so that $\sqrt{\frac{1}{n}\sum_{i\in[n]}(f(\boldsymbol{x}_i) - u_i)^2} \leq \frac{\varepsilon}{\sqrt{1/c_3^2(c_2^2/\gamma + 16)}}$. Next, we show that for any $\boldsymbol{z}_{1:n}$, the family of $\frac{(u_i - y_i)^2}{v_i} - \frac{(f^*(\boldsymbol{x}_i) - y_i)^2}{\sigma^{2*}(\boldsymbol{x}_i)} + \log\left(\frac{v_i}{\sigma^{2*}(\boldsymbol{x}_i)}\right), i \in [n]$ is an $\varepsilon$-cover for $\mathcal{H}$

$$\sqrt{\frac{1}{n}\sum_{i\in[n]}\left(\frac{(u_i - y_i)^2}{v_i} - \frac{(f^*(\boldsymbol{x}_i) - y_i)^2}{\sigma^{2*}(\boldsymbol{x}_i)} + \log\left(\frac{v_i}{\sigma^{2*}(\boldsymbol{x}_i)}\right) - \Delta_{\ell,\sigma^2,f}\right)^2}$$

$$= \sqrt{\frac{1}{n}\sum_{i\in[n]}\left(\frac{(u_i - y_i)^2}{v_i} - \frac{(f(\boldsymbol{x}_i) - y_i)^2}{\sigma^2(\boldsymbol{x}_i)} + \log\left(\frac{v_i}{\sigma^2(\boldsymbol{x}_i)}\right)\right)^2}$$

$$= \sqrt{\frac{1}{n}\sum_{i\in[n]}\left(\frac{(u_i - y_i)^2}{v_i} - \frac{(f(\boldsymbol{x}_i) - y_i)^2}{v_i} + \frac{(f(\boldsymbol{x}_i) - y_i)^2}{v_i} - \frac{(f(\boldsymbol{x}_i) - y_i)^2}{\sigma^2(\boldsymbol{x}_i)} + \log\left(\frac{v_i}{\sigma^2(\boldsymbol{x}_i)}\right)\right)^2}$$

$$\leq \sqrt{\frac{1}{n}\sum_{i\in[n]}\left(\frac{(u_i - f(\boldsymbol{x}_i))(u_i + f(\boldsymbol{x}_i) - 2y_i)}{v_i} + \frac{(f(\boldsymbol{x}_i) - y_i)^2}{\sigma^2(\boldsymbol{x}_i)} \cdot \frac{(\sigma^2(\boldsymbol{x}_i) - v_i)}{v_i} + \log\left(\frac{v_i}{\sigma^2(\boldsymbol{x}_i)}\right)\right)^2}$$

$$= \sqrt{\frac{8}{n}\sum_{i\in[n]} 1/c_3^2(c_2^2/\gamma + 16)(u_i - f(\boldsymbol{x}_i))^2 + (4c_2^2/c_3 + 1/\gamma c_3)^2(v_i - \sigma^2(\boldsymbol{x}_i))^2}$$

$$\leq 8\varepsilon.$$

Clearly, above inequality implies that $\mathcal{N}_2(\varepsilon, \mathcal{H}, \boldsymbol{x}_{1:n}) \leq \mathcal{N}_2\left(\frac{\varepsilon}{4c_2^2/c_3 + 1/\gamma c_3}, \widetilde{\mathcal{G}}, \boldsymbol{x}_{1:n}\right)\mathcal{N}_2\left(\frac{\varepsilon}{\sqrt{1/c_3^2(c_2^2/\gamma + 16)}}, \widetilde{\mathcal{F}}, \boldsymbol{x}_{1:n}\right)$. Combining the above inequality with Corollary 3.7 from Bartlett et al. (2005) we can bound the entropy number of $*\mathcal{H}$:

$$\log \mathcal{N}_2(\varepsilon, *\mathcal{H}, \boldsymbol{x}_{1:n}) \leq \log\left\{\mathcal{N}_2\left(\frac{\varepsilon}{2}, \mathcal{H}, \boldsymbol{x}_{1:n}\right)\left(\lceil\frac{2}{\varepsilon}\rceil + 1\right)\right\}$$

$$\leq \log\left\{\mathcal{N}_2\left(\frac{\varepsilon}{4c_2^2/c_3 + 1/\gamma c_3}, \widetilde{\mathcal{G}}, \boldsymbol{x}_{1:n}\right)\mathcal{N}_2\left(\frac{\varepsilon}{\sqrt{1/c_3^2(c_2^2/\gamma + 16)}}, \widetilde{\mathcal{F}}, \boldsymbol{x}_{1:n}\right)\left(\lceil\frac{2}{\varepsilon}\rceil + 1\right)\right\}.$$

Next we bound $\frac{\text{const}}{\sqrt{n}}\mathbb{E}\int_0^{\sqrt{2r^*}} \sqrt{\log \mathcal{N}_2(\varepsilon, *\mathcal{H}, \boldsymbol{x}_{1:n})}d\varepsilon$ from (36). Note that

$$\log\left\{\mathcal{N}_2\left(\frac{\varepsilon}{4c_2^2/c_3 + 1/\gamma c_3}, \widetilde{\mathcal{G}}, \boldsymbol{x}_{1:n}\right)\mathcal{N}_2\left(\frac{\varepsilon}{\sqrt{1/c_3^2(c_2^2/\gamma + 16)}}, \widetilde{\mathcal{F}}, \boldsymbol{x}_{1:n}\right)\right\}$$

$$\leq c(d_P(\widetilde{\mathcal{G}}) + d_P(\widetilde{\mathcal{F}}))\log\left(\frac{c_2^2 + 1 + c_2^2/\gamma}{\varepsilon} \cdot \frac{1}{c_3^2}\right).$$

Consequently,

$$\frac{\text{const}}{\sqrt{n}}\mathbb{E}\int_0^{\sqrt{2r^*}}\sqrt{\log\mathcal{N}_2(\varepsilon, *\mathcal{H}, \boldsymbol{x}_{1:n})}d\varepsilon$$

$$\leq \frac{\text{const}}{\sqrt{n}}\mathbb{E}\int_0^{\sqrt{2r^*}}\sqrt{\log\mathcal{N}_2\left(\frac{\varepsilon}{2}, \mathcal{H}, \boldsymbol{x}_{1:n}\right)\left(\lceil\frac{2}{\varepsilon}\rceil + 1\right)}d\varepsilon$$

$$\leq \text{const}\sqrt{\frac{(d_P(\widetilde{\mathcal{G}}) + d_P(\widetilde{\mathcal{F}}))}{n}}\int_0^{\sqrt{2r^*}}\sqrt{\log\left(\frac{1}{\varepsilon}\right)}$$

$$\leq \text{const}\sqrt{\frac{(d_P(\widetilde{\mathcal{G}}) + d_P(\widetilde{\mathcal{F}}))r^*\log(1/c_3^2 + c_2^2/c_3^2 + c_2^2/c_3^2\gamma)/r^*)}{n}}$$

$$\leq \text{const}\sqrt{\frac{(d_P(\widetilde{\mathcal{G}}) + d_P(\widetilde{\mathcal{F}}))^2}{n^2} + \frac{(d_P(\widetilde{\mathcal{G}}) + d_P(\widetilde{\mathcal{F}}))r^*\log((1/c_3^2 + c_2^2/c_3^2 + c_2^2/c_3^2\gamma)n/e(d_P(\widetilde{\mathcal{F}}) + d_P(\widetilde{\mathcal{G}})))}{n}},$$

where const corresponds to some universal constant. Together with (35) one can solve for

$$r^* \lesssim \frac{B^2C^2(d_P(\mathcal{G}) + d_P(\mathcal{F}))\log((1/c_3^2 + c_2^2/c_3^2 + c_2^2/c_3^2\gamma)n/(d_P(\mathcal{G}) + d_P(\mathcal{F})))}{n}.$$

Since $\bar{\mathbb{P}}\Delta\ell_{NLL}(\widehat{\sigma}^2, \widehat{f}; \sigma^{2*}, f^*, \boldsymbol{z}) = \mathbb{E}_{\boldsymbol{x}}\left[\frac{(f(\boldsymbol{x}) - f^*(\boldsymbol{x}))^2}{\sigma^2(\boldsymbol{x})} - 1 + \frac{\sigma^{2*}(\boldsymbol{x})}{\sigma^2(\boldsymbol{x})} - \log\left(\frac{\sigma^{2*}(\boldsymbol{x})}{\sigma^2(\boldsymbol{x})}\right)\right] \leq \frac{\varepsilon}{1/c_3^2}$, one can leverage the inequality $\log(\frac{1}{t}) \geq 1 - t + \frac{(t-1)^2}{2\max(1, t^2)}$. to conclude that

$$\mathbb{E}_{\boldsymbol{x}}\left[\left(\frac{1}{\widehat{\sigma^2}(\boldsymbol{x})} - \frac{1}{\sigma^{2*}(\boldsymbol{x})}\right)^2\right] \leq \varepsilon.$$

This completes the proof. $\qquad\square$

**Discussion** Note that the risk bound of learning the variance function using NLL loss is also studied in Zhang et al. (2023), wherein Theorem 1 suggests a rate of order $\tilde{O}\left(\frac{1}{\sqrt{n}}\right)$. On the other hand, the bound in Theorem 4.6 of this work is of the order $\tilde{O}\left(\frac{1}{n}\right)$. Such improvement of learning $\sigma^{2*}(\boldsymbol{x})$ might be of independent interest. Compared to risk bounds in Zhang et al. (2023), the major improvement comes from an application of the Local Rademacher Complexity (Bartlett et al., 2005) analysis under a Bernstein-type condition.

### A.10 Proof of Theorem 4.7 and discussion

**Proposition 1.** *Suppose Assumption 1 holds. Let $f^* \in \mathcal{F}$ and $\omega^* \in \mathcal{W}$, and suppose we have $\widehat{\omega} \in \mathcal{W}$ s.t. $\mathbb{E}_{\boldsymbol{x}}[(\widehat{\omega}(\boldsymbol{x}) - \omega^*(\boldsymbol{x}))^2] \leq \frac{\varepsilon}{b}$. Given i.i.d. samples $S_n = \{(\boldsymbol{x}_i, y_i)\}_{i=1}^n$ drawn from the data generative process, let $\widehat{f} \triangleq \arg\min_{f \in \mathcal{F}} \frac{1}{n}\sum_{\boldsymbol{z}_i \in S_n} \widehat{\omega}(\boldsymbol{x})\ell(f; \boldsymbol{z}_i)$. Then for any $\varepsilon > 0, \delta > 0$, the following holds simultaneously with probability at least $1 - \delta$:*

$$\mathbb{E}_{\boldsymbol{x}}[\widehat{\omega}^2(\boldsymbol{x})\mathcal{D}(f^*(\boldsymbol{x}), \widehat{f}(\boldsymbol{x}))] \leq \varepsilon, \quad \mathbb{E}_{\boldsymbol{x}}[\omega^{*2}(\boldsymbol{x})\mathcal{D}(f^*(\boldsymbol{x}), \widehat{f}(\boldsymbol{x}))] \leq \varepsilon, \quad \mathbb{E}_{\boldsymbol{x}}[\widehat{\omega}(\boldsymbol{x})\omega^*(\boldsymbol{x})\mathcal{D}(f^*(\boldsymbol{x}), \widehat{f}(\boldsymbol{x}))] \leq \varepsilon,$$

*as long as the sample size requirement is satisfied:*

$$n \gtrsim \frac{c_1^2 a^2(d(\mathcal{F})\log(\frac{1}{\varepsilon}) + \log(L) + \log(c_1) + \log(\frac{1}{\delta}))}{\varepsilon} + \frac{c_1 a\log(\frac{1}{\delta})}{\varepsilon}.$$

*Proof.* We start by defining the following function class:

$$\mathcal{H} \equiv \Delta \circ \omega \cdot \ell \circ \mathcal{F} \equiv \left\{\Delta\widehat{\omega}(\boldsymbol{x})\ell(f; f^*\boldsymbol{z}) = \widehat{\omega}(\boldsymbol{x})\ell(f; \boldsymbol{z}) - \widehat{\omega}(\boldsymbol{x})\ell(f^*; \boldsymbol{z}) : f \in \mathcal{F}\right\}$$

which is a composite function of hypothesis class $\mathcal{F}$, loss function $\ell$ and the difference operation $\Delta$.

To simplify the notation, we denote $\Delta_{\ell,f} = \Delta\widehat{\omega}(\boldsymbol{x})\ell(f; f^*, \boldsymbol{z})$ when there is no ambiguity. Due to the fact that $\widehat{\omega}(\boldsymbol{x})\ell(\widehat{f}; \boldsymbol{z})$ is the empirical minimizer, we have:

$$\bar{\mathbb{P}}_n\widehat{\omega}(\boldsymbol{x})\ell(\widehat{f}; \boldsymbol{z}) \leq \bar{\mathbb{P}}_n\widehat{\omega}(\boldsymbol{x})\ell(f^*; \boldsymbol{z}),$$

and thus $\bar{\mathbb{P}}_n\Delta_{\ell,\widehat{f}} \leq 0$. Leveraging such fact, next we show that $\bar{\mathbb{P}}\Delta_{\ell,\widehat{f}}$ is small via an empirical process argument. Note by assumption we have

$$\bar{\mathbb{P}}\Delta_{\ell,\widehat{f}} = \mathbb{E}_{\boldsymbol{x},y}[\widehat{\omega}(\boldsymbol{x})(\ell(\widehat{f}, \boldsymbol{z}) - \ell(f^*, \boldsymbol{z}))] = \mathbb{E}_{\boldsymbol{x}}[\widehat{\omega}(\boldsymbol{x})\omega^*(\boldsymbol{x})\mathcal{D}(f^*(\boldsymbol{x}), \widehat{f}(\boldsymbol{x}))],$$

and

$$\bar{\mathbb{P}}\Delta_{\ell,\widehat{f}}^2 - (\bar{\mathbb{P}}\Delta_{\ell,\widehat{f}})^2$$
$$= \mathbf{Var}_{\boldsymbol{z}}[\widehat{\omega}(\boldsymbol{x})(\ell(\widehat{f}, \boldsymbol{z}) - \ell(f^*, \boldsymbol{z}))]$$
$$\leq \mathbb{E}_{\boldsymbol{z}}[\widehat{\omega}^2(\boldsymbol{x})(\ell(\widehat{f}, \boldsymbol{z}) - \ell(f^*, \boldsymbol{z}))^2]$$
$$\leq \mathbb{E}_{\boldsymbol{x}}[\widehat{\omega}^2(\boldsymbol{x})\mathcal{D}(f^*(\boldsymbol{x}), \widehat{f}(\boldsymbol{x}))].$$

Since $\mathbb{E}_{\boldsymbol{x}}[(\widehat{\omega}(\boldsymbol{x}) - \omega^*(\boldsymbol{x}))^2] \leq \frac{\varepsilon}{b}$, and $\mathcal{D}(f^*(\boldsymbol{x}), \widehat{f}(\boldsymbol{x})) \leq b$ we have $\mathbb{E}_{\boldsymbol{x}}[(\widehat{\omega}(\boldsymbol{x}) - \omega^*(\boldsymbol{x}))^2\mathcal{D}(f^*(\boldsymbol{x}), \widehat{f}(\boldsymbol{x}))] \leq \varepsilon$, which implies that $\mathbb{E}_{\boldsymbol{x}}[\widehat{\omega}^2(\boldsymbol{x})\mathcal{D}(f^*(\boldsymbol{x}), \widehat{f}(\boldsymbol{x}))] \leq 2\mathbb{E}_{\boldsymbol{x}}[\widehat{\omega}(\boldsymbol{x})\omega^*(\boldsymbol{x})\mathcal{D}(f^*(\boldsymbol{x}), \widehat{f}(\boldsymbol{x}))] + \varepsilon$. To apply Lemma 1 next we find a subroot function $\psi(r)$ that

$$\psi(r) \geq 2\mathbb{E}\bar{\mathbb{P}}_n\{\Delta_{\ell,\widehat{f}} \in \mathcal{H} : \mathbb{E}[h^2] \leq r\}.$$

Note, we have $h(\boldsymbol{x}) \in [-c_1a, c_1a]$.

To find $\psi(r)$, we first analyze on the Local Rademacher Average $\mathbb{E}\mathfrak{R}_n\{\Delta_{\ell,\widehat{f}} \in \mathcal{H} : \mathbb{E}[h^2] \leq r\}$.

$$\mathbb{E}\mathfrak{R}_n(\Delta \circ \omega \cdot \ell \circ \mathcal{F}, r) = \mathbb{E}_{S_n\sigma_{1:n}}\left[\sup_{f\in\mathcal{F},\mathbb{E}_{\boldsymbol{x},y}[\Delta_{\ell,f}^2]\leq r} \frac{1}{n}\sum_{i=1}^n \sigma_i\Delta_{\ell,f}\right].$$

By Lemma 3.4 from Bartlett et al. (2005), it suffices to choose $\psi(r) \triangleq 10c_1a\mathbb{E}\mathfrak{R}_n\{*\mathcal{H}, \mathbb{P}h^2 \leq r\} + \frac{11c_1^2a^2\log n}{n}$. The following analysis largely follows from the proof in Corollary 3.7 in Bartlett et al. (2005) which aims to bound $\mathbb{E}\mathfrak{R}_n\{*\mathcal{H}, \bar{\mathbb{P}}h^2 \leq r\}$ using $\mathbb{E}\mathfrak{R}_n\{*\mathcal{H}, \bar{\mathbb{P}}_nh^2 \leq r\}$. Since $\Delta_{\ell,\widehat{f}}$ is uniformly bounded by $2c_1a$, for any $r \geq \psi(r)$, Corollary 2.2 in Bartlett et al. (2005) implies that with probability at least $1 - \frac{1}{n}$, $\{h \in *\mathcal{H} : \bar{\mathbb{P}}h^2 \leq r\} \subseteq \{h \in *\mathcal{H} : \bar{\mathbb{P}}_nh^2 \leq 2r\}$. Let $\mathcal{E}$ be event that $\{h \in *\mathcal{H} : \bar{\mathbb{P}}h^2 \leq r\} \subseteq \{h \in *\mathcal{H} : \bar{\mathbb{P}}_nh^2 \leq 2r\}$ holds, above implies

$$\mathbb{E}\mathfrak{R}_n\{*\mathcal{H}, \bar{\mathbb{P}}h^2 \leq r\} \leq \mathbb{P}[\mathcal{E}]\mathbb{E}[\mathfrak{R}_n\{*\mathcal{H}, \bar{\mathbb{P}}h^2 \leq r\}|\mathcal{E}] + \mathbb{P}[\mathcal{E}^c]\mathbb{E}[\mathfrak{R}_n\{*\mathcal{H}, \bar{\mathbb{P}}h^2 \leq r\}|\mathcal{E}^c]$$

$$\leq \mathbb{E}[\mathfrak{R}_n\{*\mathcal{H}, \bar{\mathbb{P}}_nh^2 \leq 2r\}] + \frac{2c_1^2a^2}{n}.$$

Since $r^* = \psi(r^*)$, $r^*$ satisfies

$$r^* \leq 100c_1a\mathbb{E}\mathfrak{R}_n\{*\mathcal{H}, \bar{\mathbb{P}}_nh^2 \leq 2r^*\} + \frac{50c_1^2a^2\log n}{n}. \tag{37}$$

Next we leverage Dudley's chaining bound (Dudley, 2014) to upper bound $\mathbb{E}\mathfrak{R}_n\{*\mathcal{H}, \bar{\mathbb{P}}_nh^2 \leq 2r^*\}$ using the integral of covering number.

Apply the chaining bound, it follows from Bartlett et al. (2005) Theorem B.7 that

$$\mathbb{E}[\mathfrak{R}_n(*\mathcal{H}, \bar{\mathbb{P}}_nh^2 \leq 2r^*)] \leq \frac{\text{const}}{\sqrt{n}}\mathbb{E}\int_0^{\sqrt{2r^*}} \sqrt{\log\mathcal{N}_2(\varepsilon, *\mathcal{H}, \boldsymbol{x}_{1:n})}d\varepsilon, \tag{38}$$

where *const* is some universal constant. Next we bound the covering number $\log \mathcal{N}_2(\varepsilon, \mathcal{H}, \boldsymbol{x}_{1:n})$ by $\log \mathcal{N}_2(\varepsilon, \mathcal{F}, \boldsymbol{x}_{1:n})$. We show that for all $\boldsymbol{x}_{1:n}$, any $\varepsilon/c_1 L$-cover of $\mathcal{F}$ is a $\varepsilon$-cover of $\mathcal{H}$. Specifically, let $\mathcal{V} \subset [-1,1]^n$ be an $\varepsilon/L c_1$-cover of $\mathcal{F}$ on $\boldsymbol{x}_{1:n}$ so that for all $f \in \mathcal{F}$, $\exists \boldsymbol{v}_{1:n} \in \mathcal{V}$ so that $\sqrt{\frac{1}{n}\sum_{i \in [n]}(f(\boldsymbol{x}_i) - v_i)^2} \le \frac{\varepsilon}{c_1 L}$. Now we show that for any $\boldsymbol{z}_{1:n}$, the family of $\widehat{\omega}(\boldsymbol{x}_i)(\ell(\boldsymbol{v}_i, \boldsymbol{z}_i) - \ell(f^*(\boldsymbol{x}_i), \boldsymbol{z}_i)), i \in [n]$ is an $\varepsilon$-cover for $\mathcal{H}$:

$$\sqrt{\frac{1}{n}\sum_{i \in [n]}\left(\widehat{\omega}(\boldsymbol{x}_i)(\ell(\boldsymbol{v}_i, \boldsymbol{z}_i) - \ell(f^*(\boldsymbol{x}_i), \boldsymbol{z}_i)) - \Delta_{\ell, f}\right)^2} = \sqrt{\frac{1}{n}\sum_{i \in [n]}\left(\widehat{\omega}(\boldsymbol{x}_i)(\ell(\boldsymbol{v}_i, \boldsymbol{z}_i) - \ell(\widehat{f}(\boldsymbol{x}_i), \boldsymbol{z}_i))\right)^2}$$

$$\le \sqrt{\frac{L^2}{n}\sum_{i \in [n]}\widehat{\omega}^2(\boldsymbol{x}_i)(\boldsymbol{v}_i - f^*(\boldsymbol{x}_i))^2} \le 4\varepsilon.$$

Next, combining the above inequality with Corollary 3.7 from Bartlett et al. (2005), we have the following inequality on the entropy number:

$$\log \mathcal{N}_2(\varepsilon, *\mathcal{H}, \boldsymbol{x}_{1:n}) \le \log\left\{\mathcal{N}_2\left(\frac{\varepsilon}{2}, \mathcal{H}, \boldsymbol{x}_{1:n}\right)\left(\lceil\frac{2}{\varepsilon}\rceil + 1\right)\right\} \le \log\left\{\mathcal{N}_2\left(\frac{\varepsilon}{8c_1 L}, \mathcal{F}, \boldsymbol{x}_{1:n}\right)\left(\lceil\frac{2}{\varepsilon}\rceil + 1\right)\right\}.$$

Next we bound $\frac{\text{const}}{\sqrt{n}}\mathbb{E}\int_0^{\sqrt{2r^*}}\sqrt{\log \mathcal{N}_2(\varepsilon, *\mathcal{H}, \boldsymbol{x}_{1:n})}d\varepsilon$ from (38). Note that by Haussler's covering number bound (Haussler, 1995), we have: $\log \mathcal{N}_2\left(\frac{\varepsilon}{8c_1 L}, \mathcal{F}, \boldsymbol{x}_{1:n}\right) \le cd\log\left(\frac{c_1 L}{\varepsilon}\right)$. Plugging such bound into the entropy integral yields:

$$\frac{\text{const}}{\sqrt{n}}\mathbb{E}\int_0^{\sqrt{2r^*}}\sqrt{\log \mathcal{N}_2(\varepsilon, *\mathcal{H}, \boldsymbol{x}_{1:n})}d\varepsilon \le \frac{\text{const}}{\sqrt{n}}\mathbb{E}\int_0^{\sqrt{2r^*}}\sqrt{\log \mathcal{N}_2\left(\frac{\varepsilon}{2}, \mathcal{H}, \boldsymbol{x}_{1:n}\right)\left(\lceil\frac{2}{\varepsilon}\rceil + 1\right)}d\varepsilon$$

$$\le \frac{\text{const}}{\sqrt{n}}\mathbb{E}\int_0^{\sqrt{2r^*}}\sqrt{\log \mathcal{N}_2\left(\frac{\varepsilon}{8}, \mathcal{F}, \boldsymbol{x}_{1:n}\right)\left(\lceil\frac{2}{\varepsilon}\rceil + 1\right)}d\varepsilon$$

$$\le \text{const}\sqrt{\frac{d(\mathcal{F})}{n}}\int_0^{\sqrt{2r^*}}\sqrt{\log\left(\frac{c_1 L}{\varepsilon}\right)}$$

$$\le \text{const}\sqrt{\frac{d(\mathcal{F})r^*\log(c_1 L/r^*)}{n}}$$

$$\le \text{const}\sqrt{\frac{d^2(\mathcal{F})}{n^2} + \frac{d(\mathcal{F})r^*\log(c_1 L n/ed(\mathcal{F}))}{n}},$$

where const represents some universal constant. Together with (37), one can solve for

$$r^* \lesssim \frac{c_1^2 a^2 d(\mathcal{F})\log(c_1 L n/d(\mathcal{F}))}{n}.$$

Since $\bar{\mathbb{P}}_n \Delta_{\ell, \widehat{f}} \le 0$, by Lemma 1 we have $\bar{\mathbb{P}}\Delta_{\ell, \widehat{f}} \le 2\varepsilon$ with probability at least $1 - \delta$ where the randomness comes from the training data $S_n$.

By assumption we have $\mathbb{E}_{\boldsymbol{x}}[(\widehat{\omega}(\boldsymbol{x}) - \omega^*(\boldsymbol{x}))^2] \le \frac{\varepsilon}{b}$, and $\mathcal{D}(f^*(\boldsymbol{x}), \widehat{f}(\boldsymbol{x})) \le b$ , with probability at least $1 - \delta$,

$$\mathbb{E}_{\boldsymbol{x}}[(\widehat{\omega}(\boldsymbol{x}) - \omega^*(\boldsymbol{x}))^2 \mathcal{D}(f^*(\boldsymbol{x}), \widehat{f}(\boldsymbol{x}))] \le \varepsilon,$$

which implies that

$$\mathbb{E}_{\boldsymbol{x}}[\omega^{*2}(\boldsymbol{x})\mathcal{D}(f^*(\boldsymbol{x}), \widehat{f}(\boldsymbol{x}))] \le 2\mathbb{E}_{\boldsymbol{x}}[\widehat{\omega}(\boldsymbol{x})\omega^*(\boldsymbol{x})\mathcal{D}(f^*(\boldsymbol{x}), \widehat{f}(\boldsymbol{x}))] + \varepsilon \le 3\varepsilon$$

$$\mathbb{E}_{\boldsymbol{x}}[\widehat{\omega}^2(\boldsymbol{x})\mathcal{D}(f^*(\boldsymbol{x}), \widehat{f}(\boldsymbol{x}))] \le 2\mathbb{E}_{\boldsymbol{x}}[\widehat{\omega}(\boldsymbol{x})\omega^*(\boldsymbol{x})\mathcal{D}(f^*(\boldsymbol{x}), \widehat{f}(\boldsymbol{x}))] + \varepsilon \le 3\varepsilon.$$

It can be easily verified that

$$\mathbb{E}_{\boldsymbol{x}}[\widehat{\omega}^2(\boldsymbol{x})\mathcal{D}(f^*(\boldsymbol{x}),\widehat{f}(\boldsymbol{x}))] \le 3\varepsilon \implies \mathbb{E}_{\boldsymbol{x}}[\mathcal{D}(f^*(\boldsymbol{x}),\widehat{f}(\boldsymbol{x}))|\widehat{\omega}(\boldsymbol{x}) \ge c] \le \frac{3\varepsilon}{c^2\mathbb{P}(\widehat{\omega}(\boldsymbol{x}) \ge c)}.$$

□

## B  Technical Lemmas

**Lemma 1.** *Let $\mathcal{F}$ be a class of functions ranging in $[a,b]$ and assume that there are some functional $T : \mathcal{H} \to \mathbb{R}^+$ and some constant $B$ such that for every $h \in \mathcal{H}$, $\mathbf{Var}(h) \le T(h) \le B\mathbb{P}[h] + \varepsilon$. Let $\psi$ be a subroot function and $r^*$ be the fixed point of $\psi$. Assume the $\psi$ satisfies, for any $r \ge r^*$,*

$$\psi(r) \ge B\mathbb{E}\mathfrak{R}_n\{h \in \mathcal{H} : T(h) \le r\}.$$

*Then with $c_1 = 704$ and $c_2 = 26$, for any $K > 1$ and every $t > 1$ with probability at least $1 - e^{-t}$,*

$$\forall h \in \mathcal{H}, \bar{\mathbb{P}}[h] \le \frac{K}{K-1}\bar{\mathbb{P}}_n h + \frac{c_1 K}{B}r^* + \frac{t(11(b-a) + c_2 BK)}{n} + const \cdot \varepsilon$$

*Also with probability at least $1 - e^{-t}$,*

$$\forall h \in \mathcal{H}, \bar{\mathbb{P}}_n[h] \le \frac{K+1}{K}Ph + \frac{c_1 K}{B}r^* + \frac{t(11(b-a) + c_2 BK)}{n} + const \cdot \varepsilon$$

*where $Pf = \mathbb{E}_{\boldsymbol{x}}[h(\boldsymbol{x})]$ and $\bar{\mathbb{P}}_n = \frac{1}{n}\sum_{i=1}^n h(\boldsymbol{x}_i)$.*

*Proof.* The proof is similar to the proof of Theorem 3.3 from Bartlett et al. (2005) except that here we are modifying some step so as to apply the argument under the condition $T(h) \le B\mathbb{P}h + \varepsilon$, instead of the original condition $T(h) \le B\mathbb{P}f$. We introduce notations and concepts: given class $\mathcal{H}$, $\lambda > 1$ and $r > 0$, we let $w(h) = \min\{r\lambda^k, k \in \mathbb{N}, r\lambda^k \ge T(h)\}$ and set

$$\mathcal{G}_r = \left\{\frac{r}{w(h)}h, h \in \mathcal{H}\right\}.$$

And similar to Bartlett et al. (2005) we define

$$V_r^+ = \sup_{g \in \mathcal{G}_r}\{\mathbb{P}g - \mathbb{P}_n g\}, V_r^- = \sup_{g \in \mathcal{G}_r}\{\mathbb{P}_n g - \mathbb{P}g\}.$$

Next we modify the proof step of Lemma 3.8 from Bartlett et al. (2005). Suppose $K > 1, \lambda > 0$ and $r > 0$. We aim to prove the following two claims:

$$\text{if } V_r^+ \le \frac{r}{\lambda BK} \text{ then } \forall f \in \mathcal{F}, \mathbb{P}f \le \frac{K}{K-1}\mathbb{P}_n f + \frac{r}{\lambda BK} + \frac{\varepsilon}{K-1}; \tag{39}$$

$$\text{if } V_r^- \le \frac{r}{\lambda BK} \text{ then } \forall f \in \mathcal{F}, \mathbb{P}_n f \le \frac{K+1}{K}\mathbb{P}f + \frac{r}{\lambda BK} + \frac{\varepsilon}{K}. \tag{40}$$

When $T(h) < r$, we use the same conclusion as the one in Lemma 3.8 in Bartlett et al. (2005):

$$\bar{\mathbb{P}}h \le \bar{\mathbb{P}}_n h + V_r^+ \le \bar{\mathbb{P}}_n h + \frac{r}{\lambda BK}.$$

In the case $T(h) > r$, we have $w(h) = r\lambda^k$ with $k > 0$ and $T(h) \in (r\lambda^{k-1}, r\lambda^k]$. Moreover, $g = \frac{h}{\lambda^k}$, $\mathbb{P}g \le \mathbb{P}_n g + V_r^+$ thus $\frac{\mathbb{P}h}{\lambda^k} \le \frac{\mathbb{P}_n h}{\lambda^k} + V_r^+$. Since $T(h) > r\lambda^{k-1}$, we have:

$$\mathbb{P}h \le \mathbb{P}_n h + \lambda^k V_r^+ < \mathbb{P}_n h + \frac{\lambda T(h)V_r^+}{r} \le \mathbb{P}_n h + \frac{\mathbb{P}h}{K} + \frac{\varepsilon}{K}$$

$$\implies \mathbb{P}h \le \frac{K}{K-1}\mathbb{P}_n h + \frac{\varepsilon}{K-1} + \frac{r}{\lambda BK}$$

Let $r \geq r^*$, applying Theorem 2.1 from Bartlett et al. (2005), we have for all $0 < \delta \leq 1$, with probability at least $1 - \delta$:

$$V_r^+ \leq 2(1+\alpha)\mathbb{E}\mathfrak{R}_n\mathcal{G}_r + \sqrt{\frac{2r\log(1/\delta)}{n}} + (b-a)(\frac{1}{3} + \frac{1}{\alpha})\frac{\log(1/\delta)}{n}.$$

Let $\mathcal{H}(u,v) \triangleq \{h \in MH : u \leq T(h) \leq v\}$ and define $k$ to be the smallest integer that $r\lambda^{k+1} \geq Bb + \varepsilon$. By assumption we have $T(h) \leq B\mathbb{E}[h] + \varepsilon$, and $\psi(r)$ be a sub-root function that $\phi(r) \geq B\mathbb{E}\mathfrak{R}_n\{h \in \mathcal{H} : T(h) \leq r\}$ we have:

$$\mathbb{E}\mathfrak{R}_n\mathcal{G}_r \leq \mathbb{E}\mathfrak{R}_n\mathcal{H}(0,r) + \mathbb{E}\sup_{h\in\mathcal{H}(0,Bb+\varepsilon)}\frac{r}{w(h)}\mathfrak{R}_nh$$

$$\leq \mathbb{E}\mathfrak{R}\mathcal{H}(0,r) + \sum_{j=0}^{k}\mathbb{E}\sup_{h\in\mathcal{H}(r\lambda^j,r\lambda^{j+1})}\frac{r}{w(h)}\mathfrak{R}_nh = \mathbb{E}\mathfrak{R}\mathcal{H}(0,r) + \sum_{j=0}^{k}\lambda^{-j}\mathbb{E}\sup_{h\in\mathcal{H}(r\lambda^j,r\lambda^{j+1})}\mathfrak{R}_nh$$

$$\leq \frac{\psi(r)}{B} + \frac{1}{B}\sum_{j=0}^{k}\lambda^{-j}\psi(r\lambda^{j+1}).$$

Since $\psi$ is a sub-root function, we have for all $\beta \geq 1$, $\psi(\beta r) \leq \sqrt{\beta}\psi(r)$. Hence,

$$\mathbb{E}\mathfrak{R}_n\mathcal{G}_r \leq \frac{1}{B}(1 + \sqrt{\lambda}\sum_{j=0}^{\infty}\lambda^{-j/2}).$$

Similar to Bartlett et al. (2005) we can setting $\lambda = 4$ to bound RHS by $\frac{5\psi(r)}{B}$. Since $r \geq r^* \implies \psi(r) \leq \sqrt{r/r^*}\psi(r^*) = \sqrt{rr^*}$, we have:

$$V_r+ \leq \frac{10(1+\alpha)}{B}\sqrt{rr^*} + \sqrt{\frac{2rx}{n}} + (b-a)(\frac{1}{3} + \frac{1}{\alpha})\frac{\log(1/\delta)}{n}.$$

Next we set $A = 10(1+\alpha)\frac{\sqrt{r^*}}{B} + \frac{2\log(1/\delta)}{n}$ and $C = (b-a)(1/3+1/\alpha)\log(1/\delta)/n$ so that $V+r^+ \leq A\sqrt{r}+C$. It can be verified that $r$ can be chosen such that $V_r^+ \leq \frac{r}{\lambda BK}$. The largest solution of $A\sqrt{r} + C = \frac{r}{\lambda BK}$, denoted as $r_0$. One can verify that $r_0$ is no less than $\lambda^2 A^2 B^2 K^2$ which is no less than $r^*$. Meanwhile $r_0 \leq (\lambda BK)^2A^2 + 2\lambda BKC$, by the claims in (39) and (40), one can show that for all $h \in \mathcal{H}$,

$$\mathbb{P}h \leq \frac{K}{K-1}\mathbb{P}_nh + \lambda BKA^2 + 2C + \frac{\varepsilon}{K}$$

$$= \frac{K}{K-1}\mathbb{P}h + \lambda BK(100(1+\alpha^2)\frac{r^*}{B^2} + \frac{20(1+\alpha)}{B}\sqrt{\frac{2\log(1/\delta)r^*}{n}} + \frac{2\log(1/\delta)}{n})$$

$$+ (b-a)(\frac{1}{3} + \frac{1}{\alpha})\frac{\log(1/\delta)}{n} + \frac{\varepsilon}{K}$$

Setting $\alpha = 1/10$ use the fact that $2\sqrt{uv} \leq \frac{u}{\alpha} + \alpha v$ completes the proof. $\qquad\square$

Next lemma is largely from Lemma 5.2 in Klein & Young (2015) and Zhang et al. (2024). We include the proof here for completeness.

**Lemma 2** (Chernoff type lower bound). *Let $X$ be the average of $k$ independent, 0/1 random variables (r.v.). For any $\epsilon \in (0, 1/2]$ and $p \in (0, 1/2]$, assuming $\epsilon pk \geq 6, pk \geq 6, \varepsilon \leq \frac{1}{3}$, we have:*

- *If each r.v. is 1 with probability $p$, then*

$$\mathbb{P}[X \leq (1-\epsilon)p] \geq 0.2e^{-6\epsilon^2 pk}.$$

- *If each r.v. is 1 with probability $p$, then*

$$\mathbb{P}[X \geq (1+\epsilon)p] \geq 0.2e^{-6\epsilon^2 pk}.$$

*Proof.* With Stirling's approximation, with $i!$ approximated by $\sqrt{2\pi i}(i/e)^i e^\lambda$ with $\lambda \in [1/(12i+1), 1/12i]$ one can show:

$$\binom{k}{\ell} \geq \frac{1}{e\sqrt{2\pi\ell}}\left(\frac{k}{\ell}\right)^\ell\left(\frac{k}{k-\ell}\right)^{k-\ell}. \tag{41}$$

Since $\mathbb{P}[X \leq (1-\epsilon)p] = \sum_{i=0}^{(1-\epsilon)p}\mathbb{P}[X = \frac{i}{k}]$, it suffices to provide a lower bound for $\sum_{i=(1-2\varepsilon)p}^{(1-\epsilon)p}\mathbb{P}[X = \frac{i}{k}]$, where $\mathbb{P}[X = \frac{i}{k}] = \binom{k}{i}p^i(1-p)^{k-i}$ (Klein & Young, 2015).

To this end, let $\ell = \lfloor(1-2\epsilon)pk\rfloor$. Given the fact that $\varepsilon pk \geq 6$, we have $(1-2\epsilon)pk - 1 \leq \ell \leq (1-2\epsilon)pk$. We have $\sum_{i=(1-2\varepsilon)p}^{(1-\varepsilon)p}\mathbb{P}[X = \frac{i}{k}]$ is at least

$$\varepsilon pk\mathbb{P}[X = \frac{\ell}{k}] = \frac{\varepsilon pk}{e\sqrt{2\pi\ell}}\left(\frac{k}{\ell}\right)^\ell\left(\frac{k}{k-\ell}\right)^{k-\ell}p^\ell(1-p)^{k-\ell}.$$

From Equation (41) we know that we need to bound $A = \frac{1}{e}\epsilon pk/\sqrt{2\pi\ell}$ and $B = \left(\frac{k}{\ell}\right)^\ell\left(\frac{k}{k-\ell}\right)^{k-\ell}p^\ell(1-p)^{k-\ell}$. For term $A$, since $\varepsilon pk \geq 6$, $l \leq (1-2\varepsilon)pk$ thus we need $pk \geq \frac{9e^{-2\varepsilon}}{\varepsilon}$ to get $\frac{2\varepsilon\sqrt{pk}}{e\sqrt{2\pi(1-2\varepsilon)}} \geq e^{-\varepsilon^2 pk}$. Since $\varepsilon \leq \frac{1}{3}$, it suffices to have $pk \geq 16$. To bound $B$ we need to show:

$$\left(\frac{k}{\ell}\right)^\ell\left(\frac{k}{k-l}\right)^{k-l}p^l(1-p)^{k-l} \geq e^{-4\varepsilon^2 pk}.$$

Since $\left(\frac{k}{\ell}\right)^\ell p^\ell \geq \left(\frac{1}{(1-2\varepsilon)}\right)^l$ and $(1-p)^{k-l}\left(\frac{k}{k-l}\right)^{k-l} = \left(\frac{(1-p)k}{k(1-p)+1+2\varepsilon pk}\right)^{k-l}$ we have:

$$(1-2\varepsilon)^\ell\left(1+\frac{1+2\varepsilon pk}{k(1-p)}\right)^{k-\ell} \leq e^{-\frac{4\varepsilon^2 p^2 k}{1-p}+2\varepsilon pk-2\varepsilon pk+4\varepsilon^2 pk+2} \leq 7e^{4\varepsilon^2 pk}.$$

$\square$

