# OpenReview forum: "Reweighting Improves Conditional Risk Bounds"
_TMLR — Accepted by TMLR_

### Review · Reviewer_BugY · 2024-07-08

**Summary Of Contributions:**

1)  In the paper under review, the authors study statistical properties of weighted empirical risk minimizers. They are interested in situations when reweighting helps to outperform standard ERMs. In (Zhai et al., ICLR, 2023), the authors argue that generalized reweighting algorithms lead to estimators which are very close to ERM. However, they miss the dependence on problem-dependent constants (e.g., margin). In the present paper, the authors show that weighted ERMs provably have better sample complexity compared to ordinary ERMs in several scenarios. The improvement is characterized by the factor $1/\gamma$, where $\gamma$ is a margin parameter.

2) The main result of the paper is Theorem 4.7, considering a general case. It is preceded by two particular examples: binary classification (Theorems 4.1 and 4.3) and heteroscedastic regression (Theorems 4.5 and 4.6), where reweighting naturally arises. The upper bound in Theorem 4.1 essentially matches the lower bound form Theorem 4.4. Besides, the authors show that wERM performs better than ERM in large margin regions  (Theorem 4.2).

3) The key instrument for the theoretical analysis is a weighted counterpart of the Bernstein condition called balanceable Bernstein condition. It plays essentially the same role as the standard Bernstein condition in derivation of fast rates of convergence for ERMs. However, reweighting helps to avoid distribution-dependent constants appearing in the ordinary Bernstein condtion and, as consequence, leads to tighter bounds.

**Audience:**

Yes

**Broader Impact Concerns:**

--

**Claims And Evidence:**

Yes

**Requested Changes:**

I encourage the authors to address the following comments before I can recommend acceptance.

1) Please, elaborate on implications of Theorem 4.6 in the context of distribution shift and classification with reject option.

2) Please, discuss the importance of the weighting function $\omega^*(x)$ to be bounded away from zero. Is it possible to prove similar bounds if $\omega^*$ is a $[\gamma, 1]$-valued function but the other functions from $\mathcal G$ can take values in $(0, 1]$?

**Strengths And Weaknesses:**

Strengths.

1) The paper is well-written, the literature review considers various aspects of the problem, the motivation is clear, the derivations of the main results are well-structured and easy to follow. The proof technique is quite solid though significantly relies (Bartlett et al., Ann. Statist., 2005).

2) The results show that standard ERMs, though staying minimax optimal, may be outperformed by the weighted counterparts in some subregions. This refutes the argument of Zhai et al. (ICLR, 2023) on futility of weighting schemes.


Weaknesses.

3) The main limitation of the present paper is that the authors focus on restrictive well-specified settings only. The agnostic case study is left for the future.

4) In the literature review, the authors discuss that reweighting is widely used for distribution shift adaptation and that it is also related to learning with reject option. However, the authors do not discuss how Theorem 4.6 relates to distribution shift nor compare weighted ERM with other algorithms in the context of classification with reject option.


Despite the fact that the authors consider restrictive realizable setup only, I am mostly positive about the present paper. However, I must mention that extension of the results to the agnostic setting may be extremely non-trivial and may require different techniques. The assumption that $\omega^*$ is bounded away from zero is very similar to the Massart margin condition. Fast rates of convergence for ERM under such a condition usually require $\eta^*$ to belong the  class of concepts $\mathcal F$. The only exception I am aware of is the paper of Puchkin and Zhivotovskiy (IEEE Trans. Inf. Theory, 2022), where the authors argued that the assumption $\eta^* \in \mathcal F$ is not necessary once $\mathcal F$ has finite both star number and combinatorial diameter (see Section IV). These assumptions are also quite restrictive and I am afraid that the authors will need similar requirements in the agnostic setting.


Minor remarks.

(a) Please, add references to the definitions of VC-dimension and pseudo-dimension. I think, it would be useful for those readers, who are not very familiar with learning theory.

(b) Please, describe the meaning of the inverted $\simeq$ sign (see Th. 4.4).

(c) (p.3) In the paragraph on reject option, the authors mention the paper of Klochkov and Zhivotovskiy (NeurIPS, 2021) on algorithmic stability, which is not related to prediction with reject option. Probably, the authors meant another reference.

(d) (p.7, Theorem 4.2) Misprint in the formula for $S_n$: there should be $i = 1$, instead of the subscript $i$.

(e) (p.8, Theorem 4.3) $d_P(\mathcal G) \leq \infty \quad \rightarrow \quad d_P(\mathcal G) < \infty$.

(f) (p.9, eq. (13))  Misprint: a closing bracket is missing.

---

### Review · Reviewer_ZpQh · 2024-08-06

**Summary Of Contributions:**

The aim of this paper is to propose a reweighting procedure that improves the generalization gap of standard (non-weighted) Empirical Risk Minimization (ERM). However, this improvement comes at the cost of the bound being applicable to more restricted regions of the feature space, rather than being global as before. The work is primarily theoretical, with some experimental examples on synthetic data, which I have not verified. The authors theoretically demonstrate that both the optimal weights (and thus the corresponding regions) and the new ERM estimators can be approximated for any given small error $\varepsilon$, with a significantly reduced sample complexity of only $O(1/\varepsilon)$, while still ensuring reliability (i.e., with high probability).

**Core Idea**:

This work builds upon existing variance-based bounds that leverage Bernstein-like properties, such as assuming $\mathrm{Var}(h(z))\leq B\mathbb{E}[h(z)]$ for all $h \in \mathcal{H}$, where $z$ denotes the feature-label pair $(x, y)$. In this context, each $h$ typically represents the difference between the empirical/statistical loss of a given classifier in the hypothesis set $\mathcal{F}$ and the optimal classifier, denoted by $f^*$. These bounds consist of a fast-decaying term of $O(n^{-1})$ and a slower-decaying term of $O(n^{-1/2})$, where the latter is scaled by the constant $B$. While $B$ is a constant, it can be large in some cases. To address this, the authors propose using a weight function $\omega(x)$ for the feature vector $x$ such that the inequality $\mathrm{Var}(\omega(x)h(z))\leq B'\mathbb{E}[\omega(x)h(z)]$ holds for a much smaller $B'$ compared to the original $B$. This results in a tighter bound, albeit applicable only to more specific regions of the feature space—those where the weight function $\omega(x)$ is significant.

**General Assessment**:

I have not yet verified all the proofs, but I have not identified any notable mathematical errors. The overall framework appears sound, and the results seem to be mathematically rigorous. However, the presentation of the paper leaves significant room for improvement. I have a major concern about the paper that I expect the authors to address. If this issue is resolved, I believe the paper would be a good candidate for TMLR.

**Audience:**

Yes

**Broader Impact Concerns:**

I have no concerns regarding this issue.

**Claims And Evidence:**

Yes

**Requested Changes:**

**Minor comments**:

- "can be equivalently view" -> can be equivalently viewed

- In the statement of the some theorems: What do you mean by $g\\,\mathrm{or}\\,\omega\in\mathcal{F}$? $\omega$ is the weight function and $f$ is the classifier. Do you assume they belong to a same hypothesis set?

**Strengths And Weaknesses:**

**Strengths**:

- The overall idea is novel (as far as I can tell) and interesting.

- The authors have derived several theorems to demonstrate the non-triviality of their theoretical findings. Theorem 4.1 shows that reweighting can eliminate the slow-decaying term in the generalization gap, though this result might not be entirely new (I am uncertain). However, this comes at the cost of limiting the analysis to large-margin regions. Theorem 4.2 illustrates that standard ERM has a non-decreasing lower bound everywhere, while the error of the weighted ERM can become arbitrarily small in large-margin regions given sufficient samples. Theorem 4.3 provides theoretical guarantees that the weights can be well-approximated using a relatively small number of samples. The remaining theorems, including Theorem 4.7, which offers the most general result in this paper, share a similar nature, though they vary in their assumptions and scopes.

- There are several technical discussions and remarks interspersed between the theorems that provide valuable insights to the reader.

**Weaknesses**:

- The presentation of the paper could be significantly improved. Some explanations are repeated several times within the first 4-5 pages.

- In line with the previous comment, the order in which the theorems are presented adds to the ambiguity. For example, in Theorem 4.1, the authors initially discuss a seemingly deterministic setting (where the randomness inherent in $\widehat{\omega}$ is implicit) and then suddenly assume that an inequality holds "with probability at least $1-\delta$". Probability with respect to what randomness? This becomes clearer later when Theorem 4.3 is introduced. While it's understandable that the authors want to present the main results early, this has come at the cost of readability.

- **The main weakness**: The results seem adequate, but honestly not too surprising without Theorem 4.3. Therefore, this theorem plays a crucial role in the overall technical contribution of the paper. However, there is an assumption in Theorem 4.3 that I find difficult to accept as mild or trivial. The authors assume i) knowledge of and access to a set $\mathcal{G}$ with low complexity, such as a bounded pseudo-dimension, and ii) that this set simultaneously contains the optimal weights $\eta^*$ (Bayes probabilities). This assumption facilitates the use of highly desirable bounds from the realizable setting in classical learning theory, enabling the rest of the theorems to function. However, this might be equivalent to assuming that the entire problem is being solved in a realizable setting where we already know a low-complexity hypothesis set that contains the true hypothesis. This makes the result somewhat equivalent to many prior works. I would like the authors to clarify this issue.

---

### Review · Reviewer_7fek · 2024-09-30

**Summary Of Contributions:**

This paper studies the problem of PAC learning with fast rates when the margin condition of the noise (such as Massart's noise) is not available. The paper shows that if one can estimate the margin up to a small error under the $L^2$ norm, then an empirical risk minimization weighted by the margin achieves fast rates on the same weighted population risk. The authors further demonstrate that such a guarantee on the weighted population risk implies population risk on the original loss restricted to a level set defined by the weight. In particular, this provides a PAC learner with a rejection option that outperforms bounds depending on the minimal margin. The paper then generalizes the classification scenario to regression and introduces a general condition on the weight named the "Balanceable Bernstein condition."

**Audience:**

Yes

**Claims And Evidence:**

Yes

**Requested Changes:**

Please address the points in weakness section.

**Strengths And Weaknesses:**

**Strengths**

1. The paper presents a new viewpoint for achieving fast rates that does not depend on the minimal margin.
2. The weighted ERM rule provides a natural way of abstention on hard instances, which may be useful in practice.
3. The general "Balanceable Bernstein condition" may be interesting on its own, allowing more flexibility in selecting the weight function.

**Weaknesses**

1. The main weakness, in my opinion, is that the results rely heavily on the assumption that the weight function can be estimated under the $L^2$ norm. This is questionable in practice. Although the authors mention that this can be estimated by assuming the weight function comes from a class of finite pseudo-dimension, this seems quite restrictive to me. Can the authors provide any natural hypothesis class for the weight function in specific real-world tasks? (Your experiments choose a very specific function, $0.09(1+x^2)$, which seems artificial to me.)

2. It appears to me that the technical originality is minimal, as most of the proofs boil down to local Rademacher complexity analysis. (This is not crucial given the TMLR acceptance criterion.)

3. The technical writing is not very good, especially for the proofs in the appendix. I have some specific comments below:
   - The equation (7), as stated, is unclear whether it is a consequence or an assumption. I suggest the authors clarify that it is a consequence and provide a proof in the appendix (although it is similar to the standard Massart noise case), as the condition plays a central role in the main results.
   - I suggest the authors re-define the symbols used in the proofs in the appendix to save readers from constantly checking back to the main text. For example, in the last equation on page (18), I do not understand why $\hat{f}$ suddenly becomes $f$.
   - I recommend the authors provide more intuition at the start of each proof and refrain from using words like "clearly" (as it is not always clear, especially given the typos).
   - I suggest the authors do a thorough proofreading to catch the typos, especially since the subscript for $\mathbb{E}$ is sometimes inconsistent.

---

### Author Response · Authors · 2024-12-16
**Camera-Ready Version with Code Repository Link**

We thank the Action Editor and three reviewers for their careful review of the manuscript, and for their constructive comments and suggestions.

We have uploaded the camera-ready version of the manuscript, and included the link to the code repository.

---

### Decision · Action_Editor_L2wR · 2024-11-16

**Recommendation:** Accept as is

**Comment:**

There are two core critiques of this work.

First, all reviewers argue that assuming that the weight function comes from a class of finite pseudodimension, as the authors do, is quite strong. In spite of the strength of this assumption, two reviewers do still support acceptance. In response, the authors argued that in the present era of machine learning, there are function classes over which we do learning (certain classes of deep networks, for example) that have finite pseudodimension. I think it would be tough to make the argument that we can regularly do learning over such classes, say, a decade ago; however, at this point in time, the argument is at least moderately convincing. Therefore, I am willing to accept this assumption.

The second core critique is that due to the assumptions, the analysis essentially boils down to using technical approaches that work under realizability. Moreover, it does not seem that these ideas can be extended to agnostic situations. However, given the history of results that are proved in realizability settings, and the new perspective offered by this work (by allowing for the weight function), I still do think that there the results herein are sufficiently interesting to have an audience within the learning theory community (and hence, to have some audience within TMLR). Even so, the critique stands that there do not appear to be new techniques in this work. However, it is not necessary to introduce new techniques. The novel balanceable Bernstein condition, combined with the new perspective and potential applications to classification with a reject option as well as distribution shift, make this paper interesting enough for acceptance.

I recommend this paper be accepted without minor revisions. Even so, for the camera-ready version, please do another careful pass to check for any remaining typos.

**Audience:**

Although there are concerns about the strength of the assumptions, there is sufficient audience for this work. This work provides a fresh perspective via the balanceable Bernstein condition; this condition relies on a weight function that (under a strong assumption) can be estimated. The authors arguably expand the reach of fast rates results, although they essentially are using ideas that work under realizability. With some additional work, the theory can apply to covariate shift and classification with a reject option.

**Claims And Evidence:**

All reviewers expressed sufficient confidence that the claimed theoretical results hold (i.e., there do not appear to be technical issues in the proofs). Also, the techniques used appear to be standard.